# Genome-wide association meta-analysis of human olfactory identification discovers sex-specific and sex-differential genetic variants

Franz Förster [1,2] ✉, David Emmert [3], Katrin Horn [1], Janne Pott [1,4], Johannes Frasnelli [3,5], Mohammed Aslam Imtiaz [6], Konstantinos Melas [6], Valentina Talevi [6], Honglei Chen [7], Christoph Engel [1,8], Michele Filosi [3], Myriam Fornage [9,10], Martin Gögele [3], Markus Löffler [1,8], Thomas H. Mosley [11], Cristian Pattaro [3], Peter Pramstaller [3], Srishti Shrestha [11], N. Ahmad Aziz [6,12], Monique M. B. Breteler [6,13], Kerstin Wirkner [8], Markus Scholz [1,8,14] ✉ & Christian Fuchsberger [3,14] ✉

Smelling is a human sense, expressing strong sexual dimorphisms. We aim to improve the knowledge of the genetics of human olfactory perception by performing an exploratory genome-wide association meta-analysis of up to 21,495 individuals of European ancestry. By sex-stratified and overall analysis of the identification of twelve odours and an identification score, we discovered ten independent loci, seven of them novel, with trait-wise genome-wide significance ($p < 5 \times 10^{-8}$) involving five odours. Seven of these loci, including four novel ones, are also significant using a stricter study-wide significance threshold ($p < 3.85 \times 10^{-9}$). Loci were predominantly located within clusters of olfactory receptors. Two loci were female-specific while one was sex-differential with respective candidate genes containing androgen response elements. Two-sample Mendelian randomization was applied to search for causal relationships between sex hormones, odour identification and neurodegenerative diseases. A causal negative effect was detected for Alzheimer's disease on the identification score. These findings deepen our understanding of the genetic basis of olfactory perception and its interaction with sex, prioritizing mechanisms for further molecular research.

Smell is one of the primary senses in humans. Among the human senses, olfaction is unique due to its direct neuronal connection to the amygdala, a structure of the limbic system taking part in emotional responses and memory[1]. Consequently, olfactory cues are known to activate the amygdala and can evoke autobiographical memories[2]. Odours can further be associated with emotions based on past experiences[3]. These acquired odour preferences can also influence decision making, as avoidance of negatively associated odours is typical[4].

Olfactory dysfunction often impairs quality of life[5]. It is permanently or transiently caused by sinonasal disease, viral infections of the upper respiratory tract, traumatic brain injury of varying degrees, as well as neurodegenerative diseases[6]. In patients of Alzheimer's disease (AD) or Parkinson's disease (PD), the prevalence of olfactory impairment is greater than 80%[7,8]. In PD, olfactory loss precedes the onset of the motor symptoms by several years[9] and is therefore discussed as a potential early marker[10–12].

A full list of affiliations appears at the end of the paper. ✉e-mail: franz.foerster@imise.uni-leipzig.de; markus.scholz@imise.uni-leipzig.de; Christian.Fuchsberger@eurac.edu

Odour perception exhibits pronounced sex dimorphisms. Generally, women exhibit better olfactory identification, better odour discrimination and can detect odours at lower concentrations[13,14]. Several hypotheses have been proposed to explain this phenomenon (for an overview see Sorokowski et al.)[13,15,16]. These include differences caused by gender, for example, a higher familiarity of women to culinary odours due to their traditional role in food preparation[13] and biological explanations, like changes in odour sensitivity being caused by fluctuating sex-hormone levels during the menstrual cycle. Several hormones serve as transcription regulators[17–19]. Therefore, gene-by-sex interactions may play a role in the observed sexual dimorphisms, an issue not investigated so far.

Genes involved in olfactory perception form an integral part of the human genome. Olfactory receptors (ORs), a class of G protein-coupled receptors (GPCRs), form the largest gene family in humans with around 960 genes, half of which encode functional receptors[20,21]. Besides these classical OR genes, a small number of vomeronasal receptors and trace amine-associated receptors (TAARs) contribute to olfactory perception[20].

Due to the association of olfactory dysfunction with neurodegenerative disease and the observed sexual dimorphism in olfactory performance, investigation of the genetic background of smell perception and possible interactions with sex is a worthwhile endeavour. Since the sense of smell is difficult to measure, only a few genome-wide association studies (GWAS) have been performed to date. Earlier studies have investigated isolated chemicals[22,23], whereas more recent studies[24,25] have utilised established odour test kits such as the Sniffin' Sticks screening test[26] that emulates natural odours by a combination of chemicals. To date, more than 30 variants with genome-wide significant association with different olfactory traits have been reported[22–25,27] for European American, African American, Central European and Icelandic populations. These associations were primarily found for isolated chemicals or overall olfaction rather than individual odours. Most of the reported associated variants have been located within clusters of OR genes, while others have been detected in the vicinity of genes related to neuronal function or neuronal development. The sample size of these prior studies is limited, with the largest investigating 11,326 individuals[24]. Despite the evidence for an influence of sex on olfactory perception, sex-stratified analyses were not reported in prior studies.

Here we aim to elucidate the genetic basis of human olfactory perception by performing a genome-wide association meta-analysis (GWAMA) of the ability to identify twelve common odours probed in the Sniffin' Sticks test. Our study consists of four cohorts comprising up to 21,495 subjects of European ancestry, forming the largest GWAMA of individual smell identification so far. A partial overlap with previously reported variants was observed. Additionally, we conducted sex-stratified analyses to investigate single-nucleotide polymorphism (SNP)-by-sex interactions and their biological background, uncovering a potential mechanism that could contribute to sex dimorphisms of olfaction. Using identified genetic associations, we performed two-sample Mendelian randomisation (MR) analyses to test for causal effects of sex hormones on odour identification, and to explore bidirectional causal relationships between olfaction and two major neurodegenerative diseases that are preceded by olfactory impairment, namely AD and PD.

## Results
### Genome-wide association meta-analysis
We conducted overall and sex-stratified fixed-effect GWAMA of four European ancestry GWAS to identify genetic variants associated with the identification of odour. Odour identification was measured by the Sniffin' Sticks screening test[26], which consists of a forced choice identification of twelve odour sticks given four alternatives per stick. We also analysed total odour identification performance using the sum

of correct answers across the twelve tests, i.e., the total number of correctly identified odorants, as a score. The individual odour perceptions showed no strong correlations (maximal $|r| = 0.2$, see Supplementary Fig. 1). The total sample size was 21,495 (9909 males, 11,586 females) in the identification score analysis, and 18,895 (8757 males, 10,138 females) in the 12 single odour analyses. The reduced sample size for the latter analysis is caused by incomplete recording of single odour identifications in the ARIC study. Detailed cohort descriptions are given in Supplementary Data 1. Study-wise numbers of participants with correct and incorrect odour identification are given in Supplementary Data 2.

After quality control, between 8,439,507 and 8,538,878 autosomal variants were analysed, depending on the investigated trait (see Supplementary Data 3). Quantile-Quantile (Q-Q)-plots did not show signs of genomic inflation ($\lambda$ values after quality filtering between 0.93 and 1.04) but deflation occurred in some traits, predominantly within the sex-stratified subgroups (see Supplementary Fig. 2, Supplementary Data 3). Highest heritabilities were observed for pineapple identification in females ($h^2 = 0.12$), coffee identification in females ($h^2 = 0.09$) and pineapple identification in males ($h^2 = 0.08$, see Supplementary Data 4).

According to our locus definition (see section Locus definition in Methods), eleven loci with genome-wide significant associations ($p < 5 \times 10^{-8}$) were detected (see Fig. 1) when controlling family-wise error rate (FWER) for each trait, involving associations of five of the twelve odour traits tested as well as the identification score. Seven of these loci (loci 2, 4, 5, 6, 8, 9, 11) remain significant when applying a stricter threshold of $p < 3.85 \times 10^{-9}$, controlling FWER on a study-wide level by adjusting for multiple-testing across 13 odour traits. Regional association (RA) plots of loci and forest plots of respective index variants are provided as Supplementary Figs. 3 and 4. Nominally significant heterogeneity was observed for index variants of loci 5, 8 and 11. MR-MEGA revealed nominally significant ancestry-related heterogeneity for index variants of loci 5, 7 and 9, but genome-wide significance was not affected (see Supplementary Data 5). Statistics of the index variants of the eleven loci are provided in Table 1. Loci 7, 8 and 9 were all associated with pineapple identification and are in some proximity at chromosome 11. These loci were all close to the chromosome 11 centromere, with locus 7 on the p-arm, and loci 8 and 9 on the q-arm. Respective index variants were at least 2.9 million base pairs apart. Index variants of loci 8 and 9 showed no linkage disequilibrium (LD) ($r^2 = 0$ in LDlink based on European populations[28]) and conditioning on the index variants of the other loci did not result in a loss of genome-wide significance (locus 8: $p_{cond} = 1.08 \times 10^{-18}$, locus 9: $p_{cond} = 4.21 \times 10^{-17}$). Therefore, loci 8 and 9 are considered independent. In contrast, conditioning on the index variant of locus 8 results in a loss of genome-wide significance for locus 7 ($p_{cond} = 0.63$) probably due to mild LD between the variants ($r^2 = 0.12$). Therefore, we consider this locus as conditionally dependent on locus 8, totalling the number of independent loci to ten. Conditional analysis of other loci did not reveal any additional independent associations.

### Single locus results
For each of the ten independent loci detected in our study, we annotated candidate genes based on bioinformatic annotation of variants in the 99% credible sets (CSs). Several resources were employed for this purpose, including the combined annotation dependent depletion (CADD)[29] score for evaluation of functional relevance, LD with expression quantitative trait loci (eQTLs), and previously reported GWAS associations retrieved from the GWAS catalogue. An overview of our variant-wise annotation results is provided in Supplementary Data 6 and a list of genes considered for candidate gene assignment is given in Supplementary Data 7.

We further calculated effect-direction harmonised odds ratios (Supplementary Data 8), tested index variants for interactions with sex

(SIA) (see Fig. 2, Supplementary Data 9) and performed colocalization analysis between male and female signals (see Fig. 3, Supplementary Data 9). In case of significant differences between male and female effect sizes, candidate genes were annotated by hormone response elements (HREs) (see Supplementary Data 10).

## Novel loci

The index variant of locus 1 (4q12), rs73252922, showed the strongest association with fish identification in the overall analysis ($p = 4.43 \times 10^{-8}$), with no signs of SNP-by-sex interaction ($q_{SIA} = 0.69$). The index variant achieved trait-wise but not study-wide genome-wide significance. The odds ratio was 1.71 (reference allele: A, 95% confidence interval (CI): 1.41–2.08), meaning each copy of the A allele increases the odds of identifying fish correctly by this factor. The 99% CS contained 2938 variants. Only three variants had a posterior probability (PP) above 1% reaching a cumulative PP of 41.6%. These three variants are located in the introns of *FIP1L1*. The variant with the second highest PP (13.3%) achieved a CADD value of 13.3, suggesting functional relevance. *FIP1L1* encodes a subunit of the cleavage and polyadenylation specificity factor (CPSF), which is an integral part of the polyadenylation process at the 3′ end of mRNA precursors. Two other genes near these variants were *CHIC2* and *GSX2*. CHIC2 is localised at vesicular structures, but the functional relevance remains unclear. GSX2 is a transcription factor that is part of the neurogenesis of interneurons in the olfactory bulb[30]. Due to the proximity of associated variants to *FIP1L1* and the functional relevance of *GSX2*, we consider both as plausible candidates.

At locus 2 (5p15.31) the index variant rs116058752 was associated with orange identification in females only ($p_{female} = 3.66 \times 10^{-9}$, $p_{male} = 0.57$, $p_{all} = 9.66 \times 10^{-5}$). This index variant exhibited the second strongest effect, with an odds ratio of 2.88 in females (reference allele: G, 95% CI: 2.02–4.08). The beta-beta plot of male and female effect sizes confirms that the effect was sex-specific for females ($q_{SIA} = 3.42 \times 10^{-3}$)(see Fig. 2). Likewise, in colocalization analyses of sexes there was strongest support for an association with females only (see Fig. 3). The 99% CS of the association contained 3810 variants, including three variants with a PP > 1% that combined reached a cumulative PP of 47.8%. These three variants are located within *ADCY2*, two are intronic, while the third is located in a noncoding exon. The coded protein, adenylate cyclase type 2, catalyses the second messenger cyclic adenosine monophosphate (cAMP) in the GPCR signalling pathway. Among other functions, this pathway is also relevant for OR signal transduction. Thus, *ADCY2* is a plausible candidate gene of the identified association. Notably, in contrast to ADCY3, which is directly linked to the signalling in olfactory sensory neurons, the functional role of ADCY2 is not yet fully understood[31]. A lookup of HREs for *ADCY2* found no oestrogen response elements (EREs) but four androgen response elements (AREs) full sites (best match 2bp apart from the ideal sequence motif) and five ARE half sites were described, possibly explaining the sex interaction we detected.

The index variant of locus 3 (5q21.3), rs17161232, was associated with lemon identification in the overall analysis ($p = 3.38 \times 10^{-8}$), achieving trait-wise but not study-wide genome-wide significance. The variant showed the lowest effect of all index variants with an odds ratio of 1.17 (reference allele: T, 95% CI: 1.11–1.24). No significant sex

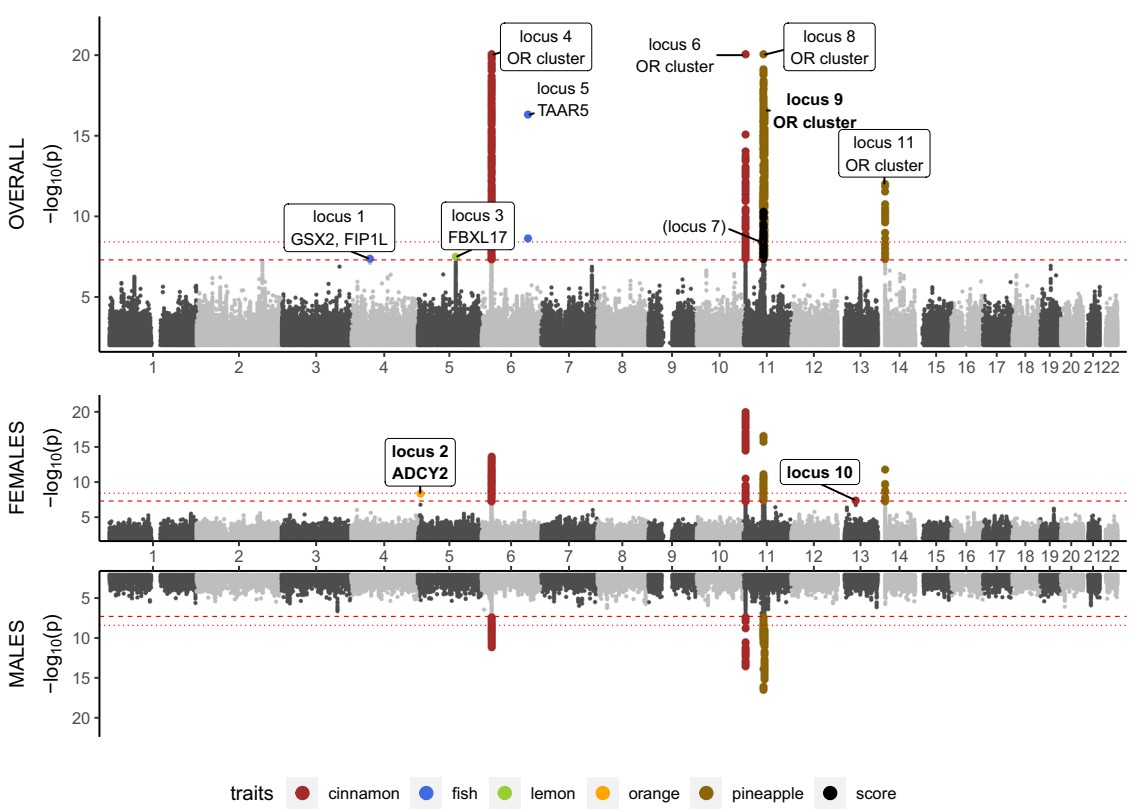

**Fig. 1 | Combined Manhattan/Miami plots of traits with genome-wide significant variants showing overall and sex-stratified association results.** The *x* axis represents chromosomal positions of variants, and the *y* axis the negative log$_{10}$-transformed *p* values from two-sided linear regression tests of association. The genome-wide significance threshold ($5 \times 10^{-8}$) is shown as a red dashed line. A stricter significance threshold ($3.85 \times 10^{-9}$) accounting for the 13 investigated traits is shown as a red dotted line. Variants with genome-wide significance are coloured according to their associated trait. Index variants are annotated with a locus identifier and proposed candidate genes, if applicable. Annotations within boxes indicate novel loci. Bold labels indicate significant sex interactions ($q_{SIA} < 0.05$). Loci in parentheses are not considered independent. Labels are positioned according to the analysis group with the strongest association of the respective index variant. SNPs with −log$_{10}$(*p*) < 2 are not shown. SNPs with −log$_{10}$(*p*) > 20 are shown with a truncated value of 20.

**Table 1 | Loci with trait-wise genome-wide significance that are associated with olfactory identification**

| Locus number | Index variant | Cytoband/Base position | Top association | AA/EA | N-weighted MAF | Beta (SE) | P value | N | Additional associated phenotypes | Proposed candidate gene(s) | q value sex interaction |
|---|---|---|---|---|---|---|---|---|---|---|---|
| 1 | rs73252922 | 4q12/53998378 | fish all | A/C | 0.0311 | −0.5391 (0.0985) | $4.43 \times 10^{-8}$ | 18632 | | FIP1L1/GSX2 | 0.69 |
| 2 | rs116058752 | 5p15.31/7705328 | orange female | A/G | 0.0293 | 1.0561 (0.1790) | $3.66 \times 10^{-9}$ | 9984 | | ADCY2 | $3.42 \times 10^{-3}$ |
| 3 | rs17161232 | 5q21.3/108318249 | lemon all | A/T | 0.1847 | 0.1589 (0.0288) | $3.38 \times 10^{-8}$ | 18678 | | FBXL17 | 0.69 |
| 4 | rs3117345 | 6p22.1/29234688 | cinnamon all | A/G | 0.4720 | −0.2587 (0.0257) | $8.14 \times 10^{-24}$ | 18690 | cinnamon female cinnamon male | OR-Cluster (OR2) | 0.74 |
| 5 | rs41286168 | 6q23.2/132589404 | fish all | A/G | 0.0126 | −1.3237 (0.1580) | $5.39 \times 10^{-17}$ | 18686 | fish all | TAAR5 | 0.14 |
| 6 | rs317787 | 11p15.4/5504313 | cinnamon all | T/C | 0.3276 | −0.3195 (0.0266) | $3.21 \times 10^{-33}$ | 18690 | cinnamon male cinnamon female | OR-Cluster (OR51/OR52) | 0.09 |
| 7* | rs11245786* | 11p11.12/50656667 | pineapple all | A/G | 0.2204 | 0.1716 (0.0291) | $3.90 \times 10^{-9}$ | 18651 | | | |
| 8 | rs669453 | 11q12.1/56493175 | pineapple all | A/C | 0.3236 | −0.2964 (0.0256) | $4.91 \times 10^{-31}$ | 18686 | pineapple female pineapple male score all | OR-Cluster (OR5) | 0.78 |
| 9 | rs61902559 | 11q12.1/59415292 | pineapple all | T/C | 0.2963 | −0.2138 (0.0253) | $2.74 \times 10^{-17}$ | 18679 | pineapple male | OR-Cluster (OR5) | 0.02 |
| 10 | rs56320200 | 13q14.3/50906784 | cinnamon female | A/C | 0.4055 | −0.1855 (0.0337) | $3.69 \times 10^{-8}$ | 10000 | | cinnamon female | $2.08 \times 10^{-5}$ |
| 11 | rs2318888 | 14q11.2/20201081 | pineapple all | T/C | 0.2818 | −0.1801 (0.0253) | $1.06 \times 10^{-12}$ | 18688 | pineapple female | OR-Cluster (OR11) | 0.13 |

Association statistics and two-sided p values of β-coefficients of the additive genetic effects are provided for the strongest association of each locus (index variant, minimum p value). The top association column reports the combination of trait and analysis group (all, male, female) corresponding to this association. Novel loci are presented in bold while significant sex interactions (two-sided) are underlined. Italicised loci fulfil a stricter study-wide significance threshold of $p < 3.85 \times 10^{-9}$, resulting from additional Bonferroni correction on the 13 investigated traits. For clusters of olfactory receptors, the most abundant OR family is provided in parentheses. For independent loci, the q values (FDR-corrected p values) for SNP-by-sex interaction of the index variant are shown. *Locus 7 was identified as not independent of locus 8 after locus definition. MAF minor allele frequency, AA alternative allele, EA effect allele, SE standard error, N sample size.

interaction was observed ($q_{SIA} = 0.69$). The index variant is in LD with multiple trait associations. The 99% CS included 152 variants. The only protein-coding gene near these variants was *FBXL17*. The gene product of *FBXL17* forms the component for substrate recognition in the SCF(FBXL17) complex, which catalyses the ubiquitylation of non-functional dimers of proteins containing a BTB binding domain to mark them for proteasomal degradation. Participation of the SCF complex in the regulation of neuronal cell fate is implicated by observations in *Xenopus laevis*[32]. The 99% CS contained 15 variants with CADD ≥ 10 of which all except two are located within *FBXL17*. We therefore consider this gene as a primary candidate of the identified association.

Three of the novel loci we detected are located within clusters of OR genes. Due to the density of olfactory-related genes, for which ligands are often unknown, we were not able to identify clear candidate genes for these loci. The index variant of locus 4 (6p22.1), rs3117345, showed its strongest association in the overall analysis of cinnamon identification ($p = 8.14 \times 10^{-24}$) without sex interaction ($q_{SIA} = 0.74$) and an odds ratio of 1.30 (reference allele: A, 95% CI: 1.23–1.36). Out of the 63 variants of the 99% CS, 11 showed a CADD score above 10. The two variants with the highest CADD scores (CADD = 23.7, respectively, 16.6) are missense mutations in an exon of *OR2J2*. The receptor OR2J2 was shown to interact with cinnamaldehyde[33], the sole component of the cinnamon stick. Another gene near variants of the 99% CS is *OR2J3*, which encodes for another receptor with reported cinnamaldehyde interaction[33,34]. A colocalization with a methylation quantitative trait locus (mQTL) annotated with *OR2J3* in the brain was observed (see Supplementary Data 11). Changes in the *OR2J3* expression level are reported for PD patients[35]. Although *OR2J2* and *OR2J3* are likely candidates, several other related genes at this locus cannot be excluded. Noteworthy, locus 4 lies in the major histocompatibility complex, which encodes for surface proteins of the immune system. Although the index variant was found to be an eQTL of HLA genes in a variety of tissues, colocalization between these signals was not supported (see Supplementary Data 11).

The index variant of locus 8 (11q12.1, rs669453) showed its strongest association with pineapple identification in the overall analysis ($p = 4.91 \times 10^{-31}$), and additionally reached genome-wide significance in both sex-stratified analyses of pineapple. The odds ratio was 1.35 (reference allele: A, 95% CI: 1.28–1.41). No SNP-by-sex interaction was observed for the index variant ($q_{SIA} = 0.78$) and colocalization of male and female signals had the strongest support (see Fig. 3). This variant was also associated at genome-wide significance with the identification score in the overall analysis, and with it, was the only variant reaching genome-wide significance for more than one trait. However, we conclude that the score association is driven by the strong pineapple association since the index variant did not even show nominal significance ($p < 0.05$) for odour traits other than pineapple and in an analysis based on the LIFE-Adult study, the score association vanishes if pineapple is excluded from the score calculation (with pineapple: $p = 5.88 \times 10^{-5}$, without pineapple: $p = 0.06$). The index variant is in strong LD ($r^2 > 0.9$) with variants reported for coffee consumption[36] and in moderate LD ($r^2 > 0.3$) with variants associated with smoking initiation[37]. However, we did not observe associations with coffee identification, indicating that variants reported for coffee consumption might play a stronger role in taste than in smell perception. This is supported by the lack of nominally significant genetic correlation between coffee identification in the overall analysis and the intake of coffee or tea (see Supplementary Data 12). The 99% CS contained 25 variants, primarily located near OR genes. However, for these receptor genes, we were not able to identify known relationships to odour components of the pineapple sniffing stick. Colocalizations of our pineapple signal and mQTLs annotated with *OR4A16*, *OR4C16* and *OR8U8* in brain were observed.

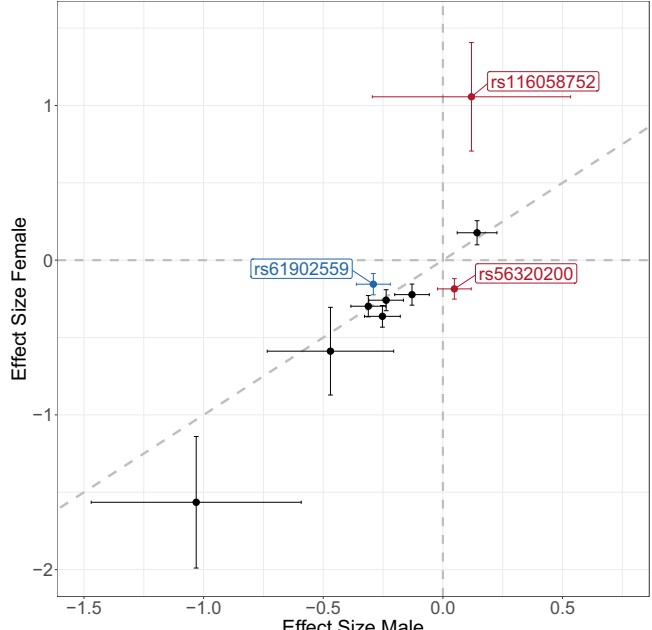

**Fig. 2 | SNP-by-sex interaction analysis of index variants of the ten independent loci.** Displayed are estimates of effect size (β-coefficients of additive genetic effects) and their respective 95% confidence limits for males (x axis) and females (y axis). Three variants achieved significance after multiple testing correction (red = higher effect size in females, blue = higher effect size in males).

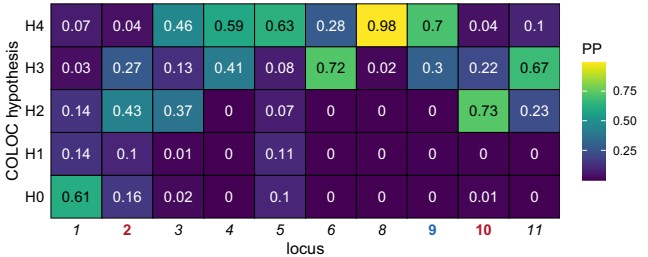

**Fig. 3 | Test for colocalization of male and female association signals at the 10 independent genome-wide significant loci.** We show the posterior probability (PP) for each of the five hypotheses. H0: no associations within the locus, H1: associations within males only, H2: associations within females only, H3: Association in both sexes but different causal variants, H4: association within both sexes with the same causal variant. Colour of the locus labels represents results of sex interaction analyses of index variants, namely red/bold: significantly higher effect in females, blue/bold: significantly higher effect in males, black/italics for loci with index variants not showing significant sex interactions.

Locus 11 (14q11.2, rs2318888) was most strongly associated in the overall analysis of pineapple identification ($p = 1.06 \times 10^{-12}$ here reaching an odds ratio of 1.20 (reference allele: T, 95% CI: 1.14–1.26). Genome-wide significant association was also observed in the subgroup of female participants, but no sex interaction was detected ($q_{SIA} = 0.13$). Of note, independent male and female signals had the strongest support in colocalization analysis (see Fig. 3). A comparison of the regional association plots between analysis subgroups at this locus is provided in Supplementary Fig. 5. The index variant in the female subgroup and the index variant in the overall analysis are in strong LD ($r^2 = 0.65$ in LDlink based on European populations[28]) but the index variant of the male subgroup is not in LD with these two variants ($r^2 = 0.085$ with the female index variant, $r^2 = 0.125$ with the index variant of the overall analysis in LDlink). Associations in sex-subgroups remained observable after conditioning on the best-

associated variant of the other sex (see Supplementary Data 13). This suggests an independent male-specific variant at this locus, which failed to reach genome-wide significance. The 99% CS of the index variant of the overall analysis contained 33 variants. Except for one of them, all variants are located within or close to OR genes. Two high CADD variants (CADD = 26.0 and 16.3, respectively) are located within *OR11H7. OR11H7* is a segregating pseudogene, and its mutations change the sensitivity to isovaleric acid[38]. Since isovaleric acid is neither a confirmed component of the pineapple stick nor of natural pineapple odour[39], it remains speculative whether this specific OR gene drives the association.

At locus 10 (13q14.3), the index variant rs56320200 was associated with cinnamon identification in females only ($p_{female} = 3.69 \times 10^{-8}$, $p_{male} = 0.18$, $p_{all} = 8.84 \times 10^{-3}$) with significant sex interaction ($q_{SIA} = 2.08 \times 10^{-5}$, see Fig. 2). The index variant showed trait-wise but not study-wide genome-wide significance. Female specificity of this locus was additionally supported by colocalization analysis (see Fig. 3). The odds ratio was 1.20 (reference allele: A, 95% CI: 1.13–1.29). Variants in LD with the index variant are associated with systolic blood pressure[40] and benign prostatic hyperplasia[41]. Chemical components of cinnamon were shown to exhibit a blood pressure-lowering effect after ingestion in diabetes patients[42], but this effect is transmitted through receptors without connection to odour perception. The 99% CS contained 89 variants located near the protein-coding genes *DLEU1, DLEU7, RNASEH2B* and *FAM124A*. None of these genes has an obvious connection to olfactory identification. All four genes were searched for HREs. One ARE full site (2 bp apart from perfect motif) was found for *RNASEH2B*, three ARE half sites for *FAM124A,* and one ARE half site for *DLEU1* (see Supplementary Data 10). Thus, no clear candidate could be identified.

## Known loci

We considered a locus as known if the index variant is in LD ($r^2 > 0.3$) with previously reported associations with olfactory traits. Three of our loci (5, 6 and 9) met this criterion. As none of the previous studies performed sex-stratified analyses, we further investigated these loci with respect to possible sex interactions. If the previously reported trait was not identical to the association observed in our study, we additionally searched for plausible candidate genes.

At locus 5 (6q23.2) the index variant rs41286168 was associated with the identification of fish in the overall analysis ($p = 5.39 \times 10^{-17}$). This variant was previously reported as an index variant by Gisladottir et al. with a genome-wide significant association for fish intensity and a suggestive association for fish naming[24]. Thus, we could confirm this association. Rs41286168 is a missense variant of *TAAR5*, which was proposed as a candidate gene. Accordingly, this variant showed the strongest effect out of all index variants, with the odds ratio reaching 3.76 (reference allele: A, 95% CI: 2.76–5.12). We observed no sex interaction for the index variant ($q_{SIA} = 0.14$), and colocalization of male and female signals had the strongest support (see Fig. 3).

At locus 6 (11p15.4), the index variant rs317787 had the strongest association with cinnamon in the overall analysis ($p = 3.21 \times 10^{-33}$) and was also significant in the corresponding sex-stratified analyses. The odds ratio was 1.38 (reference allele: T, 95% CI: 1.31–1.45). Rs317787 was previously reported as an index variant of a locus associated with cinnamon naming problems by Gisladottir et al.[24] and is located within an OR gene cluster. The signal for cinnamon identification in males colocalized with mQTLs in blood annotated with *OR51I2* and *OR52B4* (see Supplementary Data 11). A nominally significant sex interaction, not withstanding multiple testing correction, was observed ($p_{SIA} = 0.03$, $q_{SIA} = 0.09$). Colocalization analysis supported independent male and female signals (see Fig. 3). An investigation of the LD between the sex-stratified index variants showed moderate LD ($r^2 = 0.3$ in LDlink based on European populations[28], see Supplementary Fig. 6). Conditioning on the index variant of the opposite sex led to the loss of

genome-wide significance for both sexes (see Supplementary Data 13). Therefore, these variants are not considered independent.

Locus 9 (11q12.1, index variant: rs61902559) showed the strongest association for pineapple in the overall analysis and also reached genome-wide significance in males ($p_{all} = 2.74 \times 10^{-17}$, $p_{male} = 8.44 \times 10^{-16}$, $p_{female} = 9.98 \times 10^{-6}$), with an odds ratio of 1.24 (reference allele: T, 95% CI: 1.18-1.30). The index variant is in LD with a variant previously reported for β-ionone sensitivity by McRae et al.[23] (rs7943953, $r^2 = 0.76$). The near gene *OR5A1* was proposed as a candidate[43]. Because β-ionone is not a main volatile component of pineapple[39] we investigated other possible candidate genes. Three of the OR genes near the 101 variants in the 99% CS of this locus (*OR4D6*[44], *OR5A2*[45], and *OR5AN1*[46]) are known to interact with galaxolide, a component of the pineapple stick (see Supplementary Data 14). Thus, these genes may represent more plausible candidates of the observed association. The signal for pineapple identification in the overall analysis colocalized with blood mQTLs annotated with *OR10V1*, *OR4D6*, *OR5A2*, and *OR5B12* and brain mQTLs annotated with *OR5A1* and *OR5A2* (see Supplementary Data 11). Our index variant showed a sex-differential effect with larger genetic effect size in males ($q_{SIA} = 0.02$, see Fig. 2) and colocalization analysis supported a shared signal of males and females (see Fig. 3). Of the nine OR genes in a ± 250 kb window around the index variant, two (*OR4D9* and *OR4D11*) were annotated with ARE full sites (with best match 2 bp apart from ideal motif). As the ligands of these OR genes are not yet characterised, it is unclear whether they might also be plausible candidates.

### Suggestive hit analysis

We analysed loci that did not reach genome-wide but suggestive significance ($p < 10^{-6}$) in our study to get an overview of additional candidates that may influence olfactory identification but require validation in larger studies. We identified and analysed 35 suggestive loci (see Supplementary Data 15 for loci and Supplementary Fig. 7 for RA plots). We also performed SNP-by-sex interaction testing of these variants (see Supplementary Fig. 8, Supplementary Data 16).

Annotation of suggestive index variants revealed several interesting candidate genes with strong functional relation to olfactory identification (see Supplementary Data 17). Most notably, the suggestive locus 2p23.3 (associated with score in males, $p = 6.22 \times 10^{-8}$) contained *ADCY3*. The corresponding adenylate cyclase is primarily expressed in olfactory sensory cilia[47] and knock-out experiments in mice showed that a lack of ADCY3 leads to anosmia[48]. The index variant of this locus was sex-specific for males ($q_{SIA} = 1.74 \times 10^{-4}$) and multiple EREs and AREs were found for *ADCY3* (see Supplementary Data 10). Further, we identified three suggestive hits near OR genes. Two are located near OR gene clusters: locus 7q35, with the strongest association to the overall analysis of lemon ($p = 1.40 \times 10^{-7}$), and locus 12q13.2, associated with the overall analysis of liquorice ($p = 1.01 \times 10^{-7}$). Locus 13q21.31 is located near the solitary OR pseudogene *OR7E156P* and showed association in the overall analysis of pineapple ($p = 6.83 \times 10^{-7}$). None of the three index variants support SNP-by-sex interaction (7q35: $q_{SIA} = 0.19$, 12q13.2: $q_{SIA} = 0.85$, 13q21.31: $q_{SIA} = 0.48$).

### Two-sample Mendelian randomisation

Based on our GWAS findings, we investigated possible causal relationships between olfactory identification, sex hormones and neurodegenerative diseases using two-sample MR approaches. Due to the known sex dimorphisms of olfactory performance and the observed AREs near some of the identified candidate genes, we tested for a causal relationship of sex hormones, namely total testosterone (TT), bioavailable testosterone (BAT) and the hormone transporter sex-hormone binding globulin (SHBG) on smell identification. Additionally, we investigated oestradiol ($E_2$) in men.

The list of genetic instruments used for the hormones was established based on variants near genes of the steroid hormone

pathway. Hormone summary statistics were taken from Ruth et al.[49] (see Supplementary Data 18). After false discovery rate (FDR) correction on the number of exposures and the number of outcomes, no significant causal effects were detected (see Supplementary Data 19).

As olfactory impairment is associated with neurodegenerative diseases, we conducted a bidirectional two-sample MR between odour misidentification and AD and PD. We first tested whether AD (two definitions) or PD has a causal effect on the overall odour identification performance represented by the identification score. Previously reported variants for AD and PD were selected as instruments and the identification score was treated as outcome (see Supplementary Data 20). After FDR correction, we observed significant negative causal effects for all three tested exposures on the score in the overall analysis (AD wide definition: $\beta_{IVW} = -0.07$, $p_{adj} = 0.013$; AD standard definition: $\beta_{IVW} = -0.07$, $p_{adj} = 0.009$; PD: $\beta_{IVW} = -0.11$, $p_{adj} = 0.042$, see Supplementary Data 21). Beta-Beta plots of these relationships are shown in Supplementary Figs. 9–11. Wide and standard definition of AD also showed a significant negative effect on the identification score in females (AD wide definition: $\beta_{IVW} = -0.08$, $p_{adj} = 0.014$; AD standard definition: $\beta_{IVW} = -0.08$, $p_{adj} = 0.014$, see Supplementary Figs. 12 and 13). The effects of AD (wide and standard definition) on the overall score and AD (wide definition) on the score in females showed robust replication in all alternative MR methods except for Simple Median (see Supplementary Data 22). Significant MR-Egger intercepts were observed for AD (wide definition) on the score in the overall analysis, indicating bias caused by pleiotropy. Leave-one-out analysis showed that exclusion of rs1160985, a variant within the gene *TOMM40*, leads to a loss of significance for the effect of AD on the identification score, i.e., the observed causal relationship is driven by this variant (see Supplementary Figs. 14–16).

Since loss of olfaction is considered an early marker of neurodegenerative diseases, we also investigated the reverse causality direction, i.e., the causal effect of odour identification on neurodegenerative diseases. As the observed genome-wide significant association with the identification score was driven by pineapple identification rather than multiple odours, we instead considered the ability to identify single odours as exposures and AD and PD as possible outcomes. We used the index variants of the ten identified loci as instruments for olfaction (see Supplementary Data 23 for all used summary statistics). Consequently, a total of five of the twelve odours could be tested for causal relationships with AD and PD. For lemon and orange, we only had one valid instrumental variable. For cinnamon, pineapple and fish, multiple instruments were available, and hence, were combined in inverse variance weighted (IVW) analyses (see Supplementary Data 23, 24). After FDR correction, no significant causal effects were observed.

In summary, we did not find convincing support for a causal influence of sex hormones on olfactory identification or olfactory identification on neurodegenerative diseases. In contrast, AD showed robust effects on the overall odour identification performance.

We searched for additional phenotypes with a possible connection to olfaction by analysing genetic correlations between olfactory phenotypes and complex diseases. We tested the identification score in the overall analysis, as general olfaction was biologically most plausible to correlate with diseases, pineapple identification in females, the trait with the highest heritability, and coffee identification in the overall analysis because this trait showed the highest Z-score of the heritability estimate. After multiple testing corrections, no significant genetic correlations between the three olfactory traits and the considered complex diseases were observed (see Supplementary Data 25).

## Discussion

We performed a GWAMA of the identification of twelve single odours and an overall odour identification score in up to 21,495 individuals of

European ancestry. We identified ten independent loci with trait-wise genome-wide significance ($p < 5 \times 10^{-8}$) of which seven (loci 1, 2, 3, 4, 8, 10, 11) were not previously reported for olfactory traits. Seven loci, including four novel ones (loci 2, 4, 8, 11), further achieved study-wide significance ($p < 3.85 \times 10^{-9}$). All loci were exclusively associated with one specific odour. Candidate genes of these loci mostly comprise OR gene clusters but also other mechanistically plausible genes, e.g., from GPCR-based pathways and neural mechanisms. We also performed sex-stratified genetic analysis of these odour traits for the first time. Two loci showed female-specific effects, and one locus was sex-differential, with a higher genetic effect size in males. Candidate genes of these loci contained AREs, possibly explaining the interaction. MR analysis of sex hormones did not show significant causal relationships. MR revealed a causal negative effect of AD on global odour identification, while individual odour misidentification did not show a causal effect on neurodegenerative diseases.

The odotope theory, which underlies our understanding of olfactory identification, states that odour recognition is based on the activation pattern of multiple ORs and that an OR can bind multiple ligands with similar structure[50]. Therefore, a mutation in a single OR can affect the identification of multiple odours simultaneously. Our data did not comply with this expectation since none of the loci we detected were associated with multiple odours. Likewise, the single association that we observed for the identification score, intended to operationalise the impact of genetic variants on overall odour identification performance, was driven by a strong signal for pineapple detection.

Six of the ten identified loci were associated with only two odour traits, namely pineapple and cinnamon. These two odours also involve the strongest associations observed in our study and are, together with lemon, the odours showing the lowest identification performances in olfactory studies[14], increasing the power for detecting genetic associations (see Supplementary Figs. 17 and 18). Since observed odds ratios of index variants of pineapple and cinnamon were not larger than for other odours, the lower $p$ values of these associations are a result of the higher minor allele frequencies (MAFs) at these loci.

It needs to be acknowledged that the available Sniffin' Sticks test only represents a very limited excerpt of the detectable odour spectrum, which has been estimated to be larger than $1.72 \times 10^{12}$ distinguishable olfactory stimuli[51]. As the Sniffin' Sticks test is primarily designed for a clinical application, the utilised smells are primarily chosen with regard to cultural familiarity to European individuals, similar intensity and identifiability, as well as a bias towards pleasant smells[52]. Therefore, the test is not a representative selection of perceptible olfactory stimuli, as is also demonstrated in the overrepresentation of nutrition-related odours. The overlap of stick components is low, with dipentene being the only known shared major constituent of two odours, namely orange and lemon (see Supplementary Data 14). Thus, genetic effects of genes more generally involved in odour perception, like OMP, GAP43[53] or ADCY3, may have been undetected by our study and potentially require larger sample sizes and broader coverage of the human odour spectrum. This lack of olfactory coverage might also explain why no independent associations with the identification score were found.

The candidate genes identified in our study can broadly be categorised into three major groups: ORs, GPCR downstream signalling and higher neuronal function. Effects in OR clusters represent the strongest associations, suggesting that genetic factors of olfactory impairment are predominantly located within these clusters. Associations between odour identification and OR gene clusters were also reported by previous studies[22–24,27]. ORs belong to the GPCR receptor family and transmit their signals via the $G_s$ signalling pathway. This pathway also involves adenylate cyclases for the production of the second messenger cAMP, which then initiates the signalling cascade. Of note, we found two loci, one with genome-wide significance (locus

2: ADCY2) and one with suggestive significance (2p23.3: ADCY3), with assigned candidate genes coding for adenylate cyclases. Higher-order cognitive functions are generally not needed for odour perception as tested with threshold tests[54]. In contrast, the Sniffin' Sticks Screening test considered in our study was shown to be associated with cognitive function[55] and likely requires memory and verbal association abilities[54,56]. This might explain the observed associations with genes belonging to neuronal signalling or neurogenesis.

The observed sex-differential and sex-specific variants suggest that genetic mechanisms might contribute to the sex differences in human olfaction[13]. Mutations within HREs in the regulatory regions of ORs or proteins needed for their downstream signalling, like adenylate cyclases, could affect the affinity to the nuclear receptors of sex hormones. Alternatively, sex-differential expression could lead to different effects of protein sequence-modifying mutations. The presence of AREs within the vicinity of OR genes and ADCY2 makes both mechanisms viable, although these elements are only predicted by bioinformatic means without confirmation through transcription experiments[57]. We acknowledge the lack of colocalized eQTLs expected under these mechanisms, but it should be noted that expression data from olfactory bulb are limited, and with it, the ability to detect eQTLs for the most relevant tissue. Noteworthy is the lack of EREs, even though smaller studies reported that oestrogen increases olfactory performance in women[58,59]. It should be considered that this might be caused by reporting bias, as the number of reported EREs (914) and AREs (43,278 full sites) in the utilised annotation references differed by an order of magnitude. Only the suggestive ADCY3 locus, associated with the identification score in males, included EREs in addition to AREs. Associated variants of this locus are in LD with eQTLs for ADCY3. Therefore, this might be an interesting locus to investigate the impact of oestrogen on olfactory performance in future studies. In the past, cell culture-based assays have been established as a useful tool to identify odorant-receptor interactions[33,34,45,46]. To investigate our proposed mechanisms, these functional assays could be expanded by applying different levels of sex hormones and observing receptor activity or expression level. To investigate interactions further downstream in the olfactory signal processing pathway, such as the role of ADCY2 or GSX2, it is likely that in vivo models will be required. However, cross-species translations between mice and humans are not straightforward due to strong species-specific reliance on the sense of smell. This resulted in divergent evolutionary trajectories, as is demonstrated by the higher pseudogenisation rate of ORs and the loss of a functional vomeronasal organ in humans[20]. This complicates an evolutionary interpretation of the identified sex differences. Sex-differential olfactory capabilities in mice are often directly linked to mate selection and reproductive succsess[60]. In contrast, the role of olfactory cues in human mate choice and social relations is a topic of ongoing discussion.

To further clarify the possible role of hormones in the olfactory performance of the investigated odours, we performed MR analyses. As we did not observe significant causal relationships, the tested sex hormones might not have a general causal impact on olfactory identification. However, due to the high number of postmenopausal females with oestrogen levels below the detection threshold[49], we were unable to investigate the effects of oestrogen in women, although this could be a potential contributor to their observed better olfaction performance[13].

Finally, we found no support for a causal impact of olfactory impairment on neurodegenerative diseases. Conversely, a high genetic risk for AD showed a negative causal effect on overall odour identification. This suggests that although odour misidentification is a precursor of both AD and PD, it does not constitute a driving factor of these diseases. Instead, the negative effect of AD on odour identification points to an underlying shared mechanism of olfactory

impairment and AD. The associations of AD on odour identification were driven by a variant in the *TOMM40* gene. *TOMM40* codes for a translocase that participates in the import of proteins into mitochondria. Cell models have shown that *TOMM40* induces mitochondrial dysfunction, which can result in neurotoxicity[61]. Therefore, we hypothesise that neurodegeneration caused by mitochondrial dysfunction triggers both olfactory dysfunction and AD development[62]. In our primary analysis, we further observed a causal effect of PD on the identification score in the overall analysis and of AD on the score in females, but these effects were not robust regarding alternative MR methods, requiring replication in future studies. MR analyses using traits of mild cognitive impairment as a preclinical phenotype could also be of added value here.

The relatively small number of odour traits tested with the available Sniffin' Sticks tests is a limitation of our study. Testing larger sets of odours would be of high interest, but this would be costly for large-scale epidemiological studies. The high cost and complexity of olfactory testing would be a likely reason why the sample size of our study is relatively small compared to GWAS of more easily accessible traits. Due to the lower power of the sex-stratified analysis (see Supplementary Fig. 18), most of our loci were identified in the overall analysis. Our observation of several suggestive variants with plausible candidate genes indicates that larger genetic meta-analyses will be required to further advance our understanding of the genetic background of olfaction and its sex dimorphisms. Another limitation of our study arises from our focus on European individuals. Odour perception can be influenced by cultural background[63,64], perhaps a result of differing diets. It would be interesting whether this extends to genetic differences. Investigating the genetic basis of odour identification in other ethnicities is possible by adapting the Sniffin' Sticks test, as has already been done for a variety of populations, usually by changing the descriptors on the answering sheets rather than the chemical composition of the presented odours[65]. However, for trans-ethnic meta-analyses, the usage of a test independent of cultural exposure, such as odour resolution is proposed[66]. Further, we were not able to investigate gene-environment interactions. While chronic exposure to a variety of toxins has been reported to be associated with olfactory dysfunction, including heavy metals, pesticides, herbicides, solvents, or chemotherapeutic agents, there is a lack of adequate epidemiological data to perform such genetic interaction analyses[6]. These interactions might result in between-study heterogeneity, violating the assumption of our fixed-effect meta-analysis. Modelling heterogeneity with meta-regression techniques, considering structural differences between cohorts, could therefore be an interesting alternative if the number of studies and their heterogeneity increases in the future. As a final limitation, some of our proposed mechanisms of candidate genes, such as changes in general signalling by adenylate cyclases or neurogenesis, should affect general recognition of odours, while we mostly observed only associations with single odours. The suggestive hit at *ADCY3* points towards such a general mechanism, but investigation of larger numbers of odours is required to corroborate this hypothesis.

In conclusion, we added seven novel loci to the catalogue of genetic modifiers of odour identification capabilities and described sex-specific and sex-differential effects for the first time. Plausible candidate genes could be assigned to most of the loci and comprise OR genes, genes involved in GPCR downstream signalling and genes affecting brain development. No general relationships between sex hormones and odour identification were detected but a negative causal effect of AD on the overall odour identification performance. Larger studies comprising more odour traits and analysing a larger spectrum of hormones appear to be worthwhile to clarify the molecular background of the observed sex dimorphisms of human olfactory performance.

## Methods

### Contributing studies, genotyping and imputation

This GWAMA combined summary statistics of additive genetic effects on odour perception (consisting of SNP identifier, genetic position, effect allele, alternative allele, effect allele frequency, sample size, effect estimate, standard error, *p* value and imputation quality) of the following four cohorts: the Rhineland Study, CHRIS[67] (Cooperative Health Research In South Tyrol), LIFE-Adult[68] (Leipzig Research Centre for Civilisation Diseases) and ARIC[69] (Atherosclerosis Risk in Communities). Rhineland, CHRIS and LIFE-Adult are population-based studies of central European populations. ARIC contains individuals from the US with self-reported African American and European background, but only the individuals with European ancestry were included in this study due to the small sample size of the African American subgroup.

Approval to undertake the Rhineland Study was obtained from the ethics committee of the University of Bonn, Medical Faculty (reference ID: 338/15). The Rhineland Study is carried out in accordance with the recommendations of the International Conference on Harmonisation (ICH) Good Clinical Practice (GCP) standards (ICH-GCP) after the obtainment of written informed consent from all participants in accordance with the Declaration of Helsinki. The CHRIS study was conducted according to the guidelines of the Declaration of Helsinki and approved by the Ethics Committee of the Health Authority of the Autonomous Province of Bolzano (Südtiroler Sanitätsbetrieb/Azienda Sanitaria dell'Alto Adige; protocol No. 21/2011, 19 April 2011). The LIFE-Adult-Study has been approved by the ethics committee of the University of Leipzig, Germany (Reg. No. for LIFE-Adult: 263-2009-14122009 and 201/17-ek) and meets the ethical standards of the Declaration of Helsinki. Each institution of the ARIC study has Institutional Review Board approval, and each participant (or legally authorised representatives) provided written informed consent. The ARIC study complies with the Declaration of Helsinki for medical research involving human subjects.

Genotyping was performed on a per-study basis using different microarray platforms. Genotype calling, quality control and imputation were performed independently. Imputation was performed either on TOPMed[70] (CHRIS, LIFE-Adult), 1000 Genomes phase 1 version 3[71] (ARIC) or 1000 Genomes phase 3 version 5[72] (Rhineland) with standard imputation software[73–75]. For a detailed overview of study characteristics, used quality metrics, and software, see Supplementary Data 1.

### Measurement of odour identification

In the four studies, the ability of smell identification was either determined with the Sniffin' Sticks Screening 12 (Rhineland Study, LIFE-Adult, ARIC) or the Sniffin' Sticks Screening 16 test (CHRIS)[26]. During the test, participants were confronted sequentially with felt-tip pen-like odour dispensing devices containing a specific smell. For each pen, participants had to choose one of four options, with one target and three distractors. Provided options were the same across all studies, and the order of smells was consistent across individuals. In this GWAMA, the twelve odours that intersected across all four studies were investigated: orange, leather, cinnamon, peppermint, banana, lemon, liquorice, coffee, cloves, pineapple, rose and fish. Correct identification was considered as a binary trait, i.e., we combined all wrong answers into one category. We also calculated a global identification score by adding all correct answers, resulting in an integer number between zero and twelve. Pearson correlation coefficients of single odour perceptions were determined in LIFE-Adult.

### Determination of Sniffin' stick components

The lists of components of the individual Sniffin' Stick pens were extracted from the safety data sheets provided by manufacturers of the scents used in each pen to the producers of the Sniffin' Stick kits (Burghard Messtechnik GmbH) and subsequently shared with us. It

should be noted that reporting of compounds without chemical safety concerns is not mandatory. Thus, the completeness of the stick components list cannot be ensured. It should further be noted that compositions of specific pens have changed over time, further hindering the determination of the exact components (see Supplementary Data 14).

## Association analysis

Single-study association analyses were performed by the individual study analysts following a shared analysis plan. We requested to run additive genetic models with adjustment for age and current smoking status for the sex-stratified analyses and with adjustments for age, assigned biological sex and current smoking for the overall analysis. Additional adjustments were included at the discretion of the single study analysts. Accordingly, the ARIC study additionally adjusted for study centre, the Rhineland study for the first ten genetic principal components and CHRIS for pairwise genetic relatedness. Logistic regression was used for the single odour identification, and linear regression for the global identification score. The LIFE-Adult-Study and ARIC used PLINK2[76,77] for association testing, Rhineland used REGENIE[78] and CHRIS performed the analysis with SAIGE[79].

For each study, LD score regression intercepts were calculated for each trait and subgroup with LDSC (v. 1.0.1)[80] to investigate the presence of stratification bias. The default quality criteria of the tool were used for the inclusion of variants. LD scores for European populations provided by the authors of the tool were used as reference. The largest intercepts were observed for the CHRIS study, but all were < 1.1. Intercepts of the other studies were < 1.06, indicating that there is no relevant stratification bias (see Supplementary Data 26).

## Meta-analysis

Prior to meta-analysis, single study association results were harmonised by the following steps. Positions of variants from ARIC and Rhineland were lifted from hg19 to hg38 coordinates with the rtracklayer package (version 1.62.0, function liftOver)[81]. For these two studies, variants with an allele frequency deviating more than 20% from the European 1000 Genomes reference were removed, as large deviances indicate errors in genotyping or imputation. Next, monomorphic and duplicated variants were removed for all studies, as well as variants with minor allele count < 6. After harmonisation, fixed-effect meta-analysis, i.e., an analysis under the assumption that the true effect is homogeneous across studies, was performed with METAL[82] (version from 25 March 2011) using inverse-variance weighting. Between-study heterogeneity was assessed with $I^2$ statistics.

## Post-analysis quality control

We considered variants with the following properties for subsequent analyses: association statistics of the variant are available for at least two studies, sample size-weighted MAF > 1%, sample size-weighted imputation quality > 0.8 and heterogeneity $I^2 < 85\%$. Finally, variants with ambiguous allele assignment were removed.

Heritability of each trait and subgroup was estimated with LD Score regression. We used LDSC (v. 1.0.1)[80] for this purpose. SNPs were included based on the default quality criteria of the tool. The provided LD scores for European populations were used as reference.

## Visualisation of meta-analysis results

To visualise association statistics of the three analysis subgroups, males, females and overall, we constructed a combined Manhattan and Miami plot showing results of all traits with genome-wide significant findings. The R package Miamiplot[83] (version from 24 February 2021) was used for this purpose.

## Locus definition

Variants with $p < 5 \times 10^{-8}$ were considered genome-wide significant, which is consistent with controlling the FWER for each trait separately.

We also considered a stricter threshold of $p < 3.85 \times 10^{-9}$, calculated with Bonferroni correction to control the FWER study-wide by additionally accounting for the testing of 13 odour traits. Genomic loci containing genome-wide significant associations were defined across all 13 traits and three analysis groups. A locus was defined as the 1 Mb region around the variant with the lowest $p$ value across all association analyses (index variant). This procedure was iterated until all genome-wide significant associations were assigned to a locus. In case of overlapping loci, they were merged, keeping the variant with the lowest $p$ value as the index variant. The trait-group combination resulting in the lowest $p$ value was called the top association of the respective locus. Therefore, a locus can be genome-wide significantly associated with different trait-group combinations. Forest plots of index variants were created based on the R package forestplot[84]. RA plots of loci were created by using LIFE-Adult as a reference for LD and LocusZoom[85] as a reference for recombination frequencies. To derive an intuitive measure of effect, we translated the beta-coefficients of our logistic regression analyses into odds ratios using the allele representing better perception performance as reference.

## Analysis of heterogeneity

As the Sniffin' Sticks test is not highly standardised, we performed analyses of the heterogeneity. We investigated the fraction of variants with $I^2 \geq 85\%$ for the overall analysis of each trait. QC was applied prior to this analysis, but without filtering for heterogeneity. For the index variants of independent loci, we further performed a sensitivity analysis to identify studies driving heterogeneity. For this purpose, we sequentially left out a single study and recalculated the heterogeneity. It turned out that heterogeneity was not caused by a specific study, and the fraction of heterogeneous variants was similar across the investigated traits (see Supplementary Figs. 19 and 20).

To investigate the impact of study heterogeneity caused by differing ancestries, MR-MEGA regression (v.0.2)[86] was performed. Variants were not quality-filtered prior to this analysis to prevent the filtering of index variants with low MAF or imputation quality in a single study. The number of principal components used for regression was set to one, the maximal allowed value for an analysis of four studies. P values for ancestry-related heterogeneity were looked up for the index variants in their respective best-associated trait.

## Credible sets and bioinformatic annotation of variants

For every locus, we determined 99% CSs of index variants by calculating approximate Bayes factors (ABFs) for variants at the locus using the Wakefield method[87] implemented in the R-package gtx (v. 0.0.8)[88]. Priors for standard deviations were estimated based on the difference between the 97.5% and the 2.5% percentiles of the effect size distribution of the respective locus. The posterior probability (PP) for each variant was then calculated by dividing its ABF by the sum of the ABFs of all variants at the locus. Variants were ordered with decreasing PP and added to the CS until the total PP of variants within the set reached 99%.

Variants within the 99% CSs were bioinformatically annotated using various resources[89]. For compatibility with our annotation procedure and available reference data, we downlifted genetic positions from hg38 to hg19 coordinates for this purpose. Variants were annotated by Ensembl 2018[90] based gene lookup in a region of ±250 kb around the variant, CADD score[29], LD ($r^2 > 0.3$) with other GWAS variants according to the GWAS catalogue[91] (downloaded on March 8th 2024) and LD with eQTLs of the GTEx V8 catalogue (dbGaP Accession phs000424.v8.p2, downloaded on 10th June 2020)[92] as well as in-house eQTL references. LD was calculated based on 1000 Genomes Phase 3, version 5 reference panel for European populations[72]. Annotation with the GWAS catalogue was used to assess novelty. Index variants were considered unreported when they were not in LD ($r^2 > 0.3$) with previously reported variants for the respective trait.

Further, it was reported that index variants were in LD ($r^2 > 0.3$) to variants with genome-wide significant associations to other traits.

For the identification of candidate genes at novel loci, manual annotation of all genes in a ± 250 kb window of variants within the CS was performed based on GeneCards (v. 5.19)[93,94]. We then selected genes according to the following procedure: If OR genes were present at the respective locus, they were selected as candidate genes due to their direct functional relation. A literature search of all ORs at the locus level was then performed to identify known interactions with the associated odour traits. For loci without ORs, protein-coding genes with an established connection to olfaction were prioritised. Otherwise, we selected genes where a connection to olfaction was plausible based on the biological function of their gene products. If multiple plausible genes were identified, we prioritised genes near variants with high probability of being causal, high CADD values, eQTL colocalizations, and, in case of SNP-by-sex interactions, genes with hormone response elements.

## Comparison of sexes

Sex interaction analyses of all index variants of independent loci were performed according to the procedure described by Winkler et al.[95], with the extension for related individuals. Therefore, we calculated the differences between sex-stratified meta-effect estimates and standardised these differences by the respective standard errors, also taking into consideration the correlation of test statistics between sexes. The correlation of male and female effects was determined by Spearman's rank correlation of the beta-estimates of males and females. Effects were considered as sex-differential if the two-sided test of beta-estimates of males and females was significant after controlling the FDR at 5%[96]. In case of significant sex interaction and no nominal significance in one of the sexes, the effect was considered sex-specific.

## Colocalization analysis

We performed colocalization analyses for all independent loci to test whether the signals of males and females are of shared or different origins. Analyses were performed with the R package coloc (v. 5.2.3, function coloc.abf) based on Giambartolomei et al.[97]. Posterior probabilities for the following five hypotheses were computed:

- H0: No associations within the locus
- H1: Associations within males only
- H2: Associations within females only
- H3: Association in both sexes but different causal variants
- H4: Association within both sexes with the same causal variant

During candidate gene identification, we further tested for overlapping signals between identified odour associations and QTLs for different molecular traits to identify regulatory effects. As tissues, we considered blood, due to its high power, and brain tissue due to its connection to olfactory signal processing and analysed eQTLs[98,99], mQTLs[100,101] and splicing quantitative trait loci (sQTLs)[99] using data of European ancestry. QTL summary statistics for the ten independent loci with olfactory association were extracted with SMR (v1.3.1)[102] and lifted to hg38 coordinates with VCF-liftover[103]. A test for colocalization was performed if at least 50 variants intersected between the olfactory and QTL summary statistics. Colocalization between odour identification and QTLs was assumed when PP(H4) > 80%. We primarily reported identified colocalizations for candidate genes of the respective loci and genes with known olfactory function.

## Search for hormone response elements

At loci with sex-differential or sex-specific index variants, a search of genes with nearby hormone response elements (HREs) was performed. The genes were looked up in published gene lists for oestrogen response elements (EREs)[104] and androgen response elements (AREs)[57]. For the AREs the reference includes a tier system to describe deviations from the ideal motif where tier 1 contains perfect matches, tier 2 matches with 1 bp difference, tier 3 matches with 2 bp difference, tier 4 matches with 3 bp difference and tier 5 contains ARE half sites where one of the two hexamers of the ideal motif has to match perfectly.

## Conditional analysis

Conditional analyses were applied to search for additional independent variants per locus. The LIFE-Adult-Study was used as an LD reference for this purpose, as it was the largest participating cohort with accessible individual-level genotype data. For each locus, GCTA (v. 1.92.0beta3)[105] function cojo-slct was applied with forward selection. The cutoff for collinearity was set to the default value of 0.9. Conditional $p$ values below the genome-wide significance threshold of $5 \times 10^{-8}$ were considered as independent secondary hits. Conditional analysis was performed for the best associated trait.

For loci that included previously reported variants, conditional analysis (function cojo-cond) was applied, conditioning on the known variant to test whether the association found in our study was novel (independent) or reflected prior evidence. The same method was also applied to analyse the independence of neighbouring loci and to test the independence of index variants identified in sex-stratified analyses.

## Analysis of suggestive and rare variants

With the aim to identify further candidate genes and loci, we also performed locus identification, SNP-by-sex interaction testing and bioinformatic annotation of index variants reaching suggestive significance ($p < 10^{-6}$). Respective results are reported as supportive evidence requiring further validation. To reduce false positive hits, we only considered loci where at least two variants reached suggestive significance. SNP-by-sex interaction testing and annotation of index variants were performed as described for the genome-wide significant loci.

We further searched genome-wide significant associations below MAF < 1% for possible missense mutations, but no such mutations were found.

## Genetic correlation with other phenotypes

We screened for genetic correlations between common complex diseases and the overall identification score, coffee identification in the overall analysis and pineapple identification in females. The latter two traits were selected for the highest Z-score, respectively, the highest heritability estimate. Summary statistics were retrieved for AD, asthma, coronary atherosclerosis, depression, hypertension, inflammatory bowel disease, multiple sclerosis, obesity, osteoarthritis, osteoporosis, PD, rheumatoid arthritis, stroke, type 1 diabetes and type 2 diabetes from the pan-UKBB website[106]. For the genetic correlation with pineapple identification in females, sex-stratified summary statistics of these diseases were used, which, however, were not available for AD and inflammatory bowel disease. Genetic correlations were estimated with LDSC (v. 1.0.1)[80,107], using the provided LD score reference for European populations. FDR correction was performed for the number of tested phenotype pairs.

We further investigated genetic correlation between coffee odour identification and the intake of coffee and tea. Summary statistics for coffee intake and tea intake for individuals with European ancestry and were taken from Xue et al.[108].

## Mendelian randomisation analysis

Two-sample MR analysis was performed to test two major hypotheses: a causal relationship between (1) sex hormones and olfactory perception and (2) olfactory perception and neurodegenerative disorders, namely PD and two definitions of AD. The latter relationships were analysed in both directions, i.e., we performed a bidirectional MR. To select genetic instruments for hormones, we used publicly available

summary statistics for TT, BAT, SHBG and E$_2$ from Ruth et al.[49]. For summary statistics of neurodegenerative disease associations, we downloaded the meta-analysed data from FinnGen and pan-UKBB for AD (wide definition and standard) and PD[109].

In the first MR, sex hormones were considered as exposure and olfactory perception as outcome. To reduce the risk of horizontal pleiotropy, i.e., effects not mediated by the exposure, as best as possible, we restricted the list of instruments to variants in regions of ±250 kb around genes of the steroid hormone pathway as defined in KEGG (hsa00140)[110]. The 62 genes are located in 34 unique cytobands, and for each cytoband and hormone, we filtered for a significant signal (abs($Z$) > 5.45) in either men, women, or the overall analysis. In our primary MR analysis, we used the resulting list of strong instruments of the respective analysis groups (male, female, overall). Resulting summary statistics are given in Supplementary Data 18. To allow for comparison between the sexes, we performed a sensitivity analysis based on the common variant list for both sexes per hormone. For this, we tested for colocalization between men and women[97] (see Supplementary Data 27). In case of positive colocalization (PP(H4) > 0.5), the best-associated variant of either men or women was selected as representative for the locus and both sexes. In case of different signals (PP(H3) > 0.5), we kept both variants as independent instruments at the same cytoband. In case of a sex-specific signal (PP(H1) > 0.5 or PP(H2) > 0.5), the best-associated variant for both sexes was selected, even if it was a weak instrument for one of the sexes. The variant is discarded if none of these conditions are met. In a second sensitivity analysis, we restricted instruments to strong variants colocalising between males and females.

To test the causal effects, we performed a step-wise approach: first, we screened all hormone-odour combinations in their matching analysis group using the MR-IVW random effects model[111,112] as implemented in the R-package MendelianRandomization (v. 0.10.0)[113]. Multiple-testing correction was performed for each sex-setting separately with a two-step procedure. At the first step, an FDR correction was performed for each odour outcome over the number of hormone exposures using the Benjamini & Hochberg procedure[96]. In the second step, the adjusted $p$ values from the first step were FDR corrected per exposure on the number of outcomes tested. We then selected all combinations with multiple testing corrected $p$ values < 0.05 and tested them again using MR-LOO, MR-Egger[114], median-based estimates[115], and MR with penalised weights[116]. A combination of hormone and stick was considered significant if the adjusted $p$ value of the IVW-analysis was < 0.05 and the results could be replicated using the other MR methods (nominally significant and same effect direction).

To investigate the bidirectional causal relationship between olfaction and neurodegenerative diseases, we first looked at the influence of neurodegenerative diseases on the odour identification score. Genome-wide significantly associated variants of AD (wide and standard definition) or PD with low heterogeneity in the meta-analysis of FinnGen and UKB were selected as instruments. We performed harmonisation of effect alleles by changing the sign of the effect for variants with inverted alleles. Then we performed position-based priority pruning to obtain a list of independent variants. In short, we sorted the SNP list by $p$ value and excluded variants of lower rank when their position was within ±1 Mb of the higher-ranking SNP. The odour identification score of the sex-combined analysis was used as the outcome in the primary analysis, while the sex-stratified scores were used as sensitivity analyses. For the reverse direction, i.e., the influence of odour detection on neurodegenerative diseases, we used the identified ten independent genome-wide significant variants as instruments of their corresponding best-associated odour trait and analysis group. As sensitivity analysis, we considered the test statistics of the other analysis groups of the corresponding trait, if they reached genome-wide significance as well. For the reverse causal relationship of olfaction on neurodegenerative diseases, we first estimated the

Wald ratio estimate[117] per variant and outcome using the first two terms of the delta method expansion to estimate the variance of the ratio. For forward and reverse direction, we then performed the MR-IVW fixed effect model[111] when multiple variants for an exposure were available. FDR was again performed with a two-step approach for forward and reverse direction by adjusting for the number of exposures in the first step and the number of outcomes in the second step for each mode and sex-setting. Combinations with significant IVW effect were then reanalysed with MR-LOO, MR-Egger[114], median-based estimates[115], and MR with penalised weights[116].

## Power analysis

Power analysis to evaluate the impact of different Sniffin' Stick-wise case rates and study sample sizes was performed with the function genpwr.calc from the R library genpwr. Case rates were determined across participating studies by counting the stick-wise frequency of odour misidentification. Power calculation was performed under the assumption of the correctness of the additive genetic model. Covariate effects were not considered.

## Reporting summary

Further information on research design is available in the Nature Portfolio Reporting Summary linked to this article.

# Data availability

Summary statistics for this study have been deposited in the Leipzig Health Atlas under accession code 8VK7H50F6P-6 (https://www.health-atlas.de/assays/88). Further data are provided in the Supplementary Data file. The raw individual-level data are protected and are not publicly available due to data privacy laws, but can be requested from individual studies. Data from the Rhineland study can be requested by researchers in accordance with the Rhineland Study's Data Use and Access Policy (RS-DUAC@dzne.de). Access to CHRIS data can be provided for research purposes upon request to the CHRIS Access Committee (access.request.biomedicine@eurac.edu). Data from the LIFE-Adult-Study is available based on written project agreements and data transfer agreements according to the centre's data use and access policies[68]. ARIC data from visit 1 to visit 5 are available through the Biologic Specimen and Data Repository Information Coordinating Centre (BioLINCC). Data that are not yet available through BioLINCC are available upon request through the ARIC Coordinating Centre at the University of North Carolina. Data sets used in the study are Ensembl 2018 (http://www.ensembl.org/index.html), CADD (https://cadd.gs.washington.edu/download), GTEx V8 (https://gtexportal.org/home/protectedDataAccess), pan-UKBB (https://pan.ukbb.broadinstitute.org/downloads/), FinnGen (https://www.finngen.fi/en/access_results) and publicly available summary statistics for sex hormones (https://www.ebi.ac.uk/gwas/publications/32042192), xQTLs (https://yanglab.westlake.edu.cn/software/smr/#DataResource), as well as intake of coffee and tea (https://yanglab.westlake.edu.cn/pub_data.html).

# Code availability

All analysis scripts are available on Zenodo (https://doi.org/10.5281/zenodo.15606619)[118].

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

## Acknowledgements

We thank Eva König for her assistance with the colocalization analysis. The LIFE-Adult-Study is part of the Leipzig Research Centre for Civilisation Diseases, LIFE. LIFE is an organisational unit of the Medical Faculty of the University of Leipzig, Germany. LIFE was funded by means of the European Union, by the European Regional Development Fund (ERDF) and by funds of the Free State of Saxony within the framework of the Saxonian excellence initiative. We wish to thank the citizens of the City of Leipzig for their willingness to participate in the study. CHRIS Study investigators thank all study participants, the Healthcare System of the Autonomous Province of Bolzano-South Tyrol, and all Eurac Research staff involved in the study (full CHRIS acknowledgements can be found here: https://www.eurac.edu/en/institutes-centers/institute-for-biomedicine/pages/acknowledgements) Bioresource Impact Factor Code: BRIF61. The CHRIS Study was funded by the Autonomous Province of Bolzano-South Tyrol–Department of Innovation, Research, University and Museums and supported by the European Regional Development Fund (FESR1157). The Rhineland Study is supported by DZNE core funding at the German Centre for Neurodegenerative Diseases (DZNE). This work was further supported by the Federal Ministry of Education and Research (BMBF) as part of the Diet-Body-Brain Competence Cluster in Nutrition Research (grant number 01EA1410C) and in the framework of "PreBeDem - Mit Prävention und Behandlung gegen Demenz" (FKZ: 01KX2230). The Atherosclerosis Risk in Communities Study is carried out as a collaborative study supported by National Heart, Lung, and Blood Institute contracts (75N92022D00001, 75N92022D00002, 75N92022D00003, 75N92022D00004, 75N92022D00005). The ARIC Neurocognitive Study is supported by U01HL096812, U01HL096814, U01HL096899, U01HL096902, and U01HL096917 from the NIH (NHLBI, NINDS, NIA and NIDCD). Funding was also supported by R01HL087641 and R01HL086694; National Human Genome Research Institute contract U01HG004402; and National Institutes of Health contract HHSN268200625226C. Infrastructure was partly supported by Grant Number UL1RR025005, a component of the National Institutes of Health and NIH Roadmap for Medical Research. The authors thank the staff and participants of all the participating cohorts for their important contributions. We also want to acknowledge the participants and investigators of the FinnGen study. This project was funded by the Deutsche Forschungsgemeinschaft (DFG, German Research Foundation) (SFB-1052/4 B11 to M.S., F.F.), the ministry for science and health of the Rhineland-Palatinate (CoAGE graduate programme, F.F.), the German Federal Ministry of Education and Research (BMBF) (grant #01ZX1906B, project SYMPATH, K.H.), the BMBF and the federal states of Germany and the Helmholtz society (grant # 01ER2301/14, NAKO, K.H.), the United Kingdom Research and Innovation Medical Research Council (MC_UU_00002/7, J.P.), the Wellcome Trust (225790/Z/22/Z, J.P.), an Alzheimer's Association Research Grant (Award Number: AARG-19-616534, N.A.A.), a European Research Council Starting Grant (101041677, N.A.A.), and the NIH (contract 75N92022D00004, T.H.M.). Computations were performed using the IT infrastructure of the Centre for Scalable Data Analytics and Artificial Intelligence (ScaDS.AI) Dresden/Leipzig, funded by the German Federal Ministry of Education and Research (BMBF grant #01IS18026B). Publishing was supported by the Open Access Publishing Fund of Leipzig University.

## Author contributions

F.F.: formal analysis, writing—original draft, D.E.: analysis of CHRIS data, meta-analysis, writing—methodology and critical review, K.H.: formal

analysis, preparation of life-adult-study data, writing–critical review, J.P.: Mendelian randomisation, writing—methodology and results, J.F.: research of stick components, writing—critical review, M.A.I.: analysis of Rhineland data, writing—critical review, K.M., V.T., H.C., M.Fi., M.Fo., M.G., M.L., T.H.M., C.P., P.P., S.S., N.A.A., M.M.B.B., writing—critical review, C.E.: participation in conceptualisation, preparation of LIFE-Adult-Study data, writing—critical review K.W.: preparation of LIFE-Adult-Study data, writing—critical review, M.S. and C.F.: conceptualisation, writing—manuscript editing and critical review, supervision.

## Funding

## Competing interests
M.S. received funding from Owkin for a project not related to this research. The remaining authors declare no competing interests.

## Additional information

[1]Institute for Medical Informatics, Statistics and Epidemiology (IMISE), Medical Faculty, Leipzig University, Leipzig, Germany. [2]Institute of Molecular Biology gGmbH (IMB), Mainz, Germany. [3]Institute for Biomedicine, Eurac Research, Bolzano, Italy. [4]University of Cambridge, MRC Biostatistics Unit, Cambridge, United Kingdom. [5]Department of Anatomy, Université du Québec à Trois-Rivières, Trois-Rivières, Québec, Canada. [6]German Centre for Neurodegenerative Diseases (DZNE), Population Health Sciences, Bonn, Germany. [7]Department of Epidemiology and Biostatistics, Michigan State University, East Lansing, MI, USA. [8]LIFE - Leipzig Research Centre for Civilization Diseases, Medical Faculty, Leipzig University, Leipzig, Germany. [9]Brown Foundation Institute of Molecular Medicine, McGovern Medical School, University of Texas Health Science at Houston, Houston, TX, USA. [10]Human Genetics Center, School of Public Health, University of Texas Health Science at Houston, Houston, TX, USA. [11]The Memory Impairment and Neurodegenerative Dementia (MIND) Center, University of Mississippi Medical Center, Jackson, MS, USA. [12]Department of Neurology, Faculty of Medicine, University of Bonn, Bonn, Germany. [13]Institute for Medical Biometry, Informatics and Epidemiology (IMBIE), Faculty of Medicine, University of Bonn, Bonn, Germany. [14]These authors jointly supervised this work: Markus Scholz, Christian Fuchsberger. ✉e-mail: franz.foerster@imise.uni-leipzig.de; markus.scholz@imise.uni-leipzig.de; Christian.Fuchsberger@eurac.edu

