## [Transparent Peer Review file · Nature Communications]

Genome-wide association meta-analysis of human olfactory identification discovers sex-specific and sex-differential genetic variants

Corresponding Author: Mr Franz Förster

Version 0:

Reviewer comments:

Reviewer #1

(Remarks to the Author)

This study investigated the genetics of human olfactory perception through a comprehensive genome-wide association meta-analysis involving over 21,000 individuals of European ancestry. By utilizing a standardized smell test called the 'Sniffin' Sticks,' researchers examined the ability to identify 12 different odours. They identified ten genetic regions linked to specific scents, with seven being newly discovered. Unique to this work, the authors conducted analyses based on sex, finding loci that were either female-specific or showed stronger genetic effects in males. Many of the implicated genes were related to olfactory receptors or pathways involved in cell signalling and neural function. Additionally, the study used Mendelian Randomization techniques to reveal a potential causal link between bioavailable testosterone and the identification of a specific scent (pineapple) in men, as well as a connection between Alzheimer's disease and general olfactory abilities.

The findings provide insight into how genetics may shape differences in smell perception between men and women and suggest a genetic basis for the observed sexual dimorphism in olfaction. The study also investigates connections between olfactory function and neurodegenerative diseases such as Alzheimer's and Parkinson's, with implications for early diagnosis and understanding of disease progression. However, the researchers note limitations, such as a relatively narrow range of scents tested and a sample restricted to European ancestry, pointing to the need for broader, more diverse studies to fully capture the genetic landscape of smell perception.

Major comments

- 1) The introduction could be expanded by discussing olfactory perception within the broader context of sensory and cognitive functions. Exploring its connections to memory, decision-making, and emotional responses would emphasize the multifaceted importance of this sense and its relevance to human behaviour and health.
- 2) A brief mention of other health conditions linked to olfactory dysfunction, such as metabolic disorders, cancer, or infectious diseases, could widen the paper's appeal and emphasize the broad significance of olfactory research.
- 3) The reliance on the 'Sniffin' Sticks' test for the measurement of olfactory traits, might limit the breadth of olfactory traits examined. It might be worth discussing how this choice may have affected the scope of the study. Also, consider discussing why these particular smells were chosen.
- 4) Provide a discussion on how populations of different ancestry could drastically change the outcome of the study, as some 'smells' might be completely foreign to them. For future studies, how should one select the smells to test differences in olfaction, based on the demography? And how one might be able to universalise these effects.
- 5) In the methods, the association analysis uses the genome-wide threshold of $5e-8$. As a separate test is effectively conducted for each of the 13 odour loci, this significance threshold should be adjusted to account for multiple testing. This may alter the number of significant (novel) loci identified (depending on the correction method used). For example, a rough statistical bon ferroni adjustment would shift the significance threshold such that 2 of the novel loci would no longer reach genome-wide significance.
- 6) Since six loci were found to be associated with pineapple and cinnamon, it would be useful to provide more information on the specificity of the associations. Discuss on why these particular odours might be more strongly linked genetically.

- 7) Providing more context about the biological relevance and magnitude of effect sizes (e.g., how meaningful these effects are in practical terms) would be helpful. This is particularly relevant for loci showing strong associations with specific odours like pineapple or cinnamon.
- 8) The mechanistic claims between the genetic loci, sex hormones, and olfaction are speculative. It would benefit by discussing past literature if available on functional assays of these mechanistic elements and how they relate to sex-specific olfactory responses. Providing some future directions on how such functional assays could be performed to understand these effects would be beneficial to the wide readership.
- 9) Since the authors did not find strong evidence for neurodegenerative diseases it might be worth reframing the introduction in a way that does not emphasize too heavily on AD and PD.
- 10) Mention and discuss how environmental factors such as diet, exposure to pollutants, and cultural habits can interact with genetic predispositions to influence olfactory function.
- 11) It would be worthwhile to provide an evolutionary perspective on why olfactory traits might exhibit strong sexual dimorphisms and how this might relate to survival, reproduction, and social interactions in humans can enrich the narrative and draw connections to evolutionary biology.
- 12) The identification of ten independent loci, including seven novel ones, is an important finding, the results could benefit from more in-depth explanations about why these loci may have been missed in previous studies. This context would help clarify the novelty and importance of the findings.
- 13) The causal relationship between bioavailable testosterone and pineapple odour identification in males needs context on why this specific relationship might exist is needed. Are there known links between testosterone and olfactory pathways that make this result plausible?

Minor comments

Line 56-62: This paragraph would be better placed as the first and then one introduces the neurodegenerative diseases followed by the genes paragraph..this would flow better with the paragraph starting line 70.

Line 57: What does "lower detection threshold" mean

Line 57-58: It would be better to elucidate a couple of these hypotheses, it reads a bit dry as it is framed now

Line 60-62: This sentence reads truncated. Consider expanding and rephrasing or making two separate sentences.

Line 80: Reframe, as in the current study the sex-stratified analysis actually has fewer number of individuals. Suggesting the current analysis is also limited as this ref 19 one.

Line 81: It would be good to mention the demographics of the populations used in these studies and how the current study in the following paragraph strengthens the existing literature

Line 90: This is a big claim. The study hasn't really found extensive evidence to support this claim. Consider rephrasing

Line 92-93: Expand on the bidirectional relationship tested. This phrasing makes it sound like the study only addressed the relationship between odour on disease and not vice-versa (wherein the novel loci were found)

Line 101-103: Explain the discrepancy in the sample sizes for the two analyses

Line 210-211: Consider expanding on this result in the discussion, as it is a one-off comment at this stage

Line 240-241: Might be interesting to link this information with cinnamon specificity in females, in the discussion. See <https://pubmed.ncbi.nlm.nih.gov/23867208/> for more details

Line 352-354: Discuss the potential mechanisms (e.g., impact on olfactory bulbs or neurodegeneration). Further, clarifying why no similar effect was found for Parkinson's disease may help explain potential differences in disease pathology.

Line 358-359: It would be helpful to focus on/remind the reader which ones were novel

Line 373-376: Speculative claim. Consider providing evidence for this.
In addition, it would be good to provide information in the supplementary on the identification rate per smell

Line 378-382: Were these odours chosen due to their strong genetic effects, or do they reflect biases in test composition? Addressing this would help readers understand potential limitations in trait selection.

Line 387: Consider mentioning "only known" as the composition mentioned previously seems to not have the complete information

Line 388: This needs a reference, what studies and what genes are involved, need to be mentioned.

Line 389-390: It might be more nuanced than this, consider rephrasing the explanation
Line 407-408: Overstating as only two variants were found

Line 428: Mention that this is only for this context of the study, the smells tested here. It might not be a universal statement

Line 471: elaborate on the summary stats

Line 476: What is the threshold for European ancestry? Bi or uniparental and what is the time cut-off

Line 494-496: Is this accounted for in the analysis, as this might inflate results and/or introduce artefacts

Line 505-506: Was the order of the smells the same or not for each individual

Line 506-508: Is there a way to provide information on how the identification of each smell was correlated with the other smells? For example, does identifying pineapple mean one never identifies rose etc? Would be helpful to state this somewhere

Line 517-518: Mention which pens, and across which studies. Especially relevant for pineapple and cinnamon

Line 521: Why not interactive models, as one would lose information otherwise

Line 524-525: but no control of genetic relatedness?

Line 526: Pairwise genetic relatedness? This seems quite vague.

Line 531-532: Quite a bit vague - what does "harmonising" entail? Would not be able to replicate this analysis based on this description. Does it control for the different microarrays/ imputation approaches or how the different studies account (or do not account) for shared ancestry?

hg19 to hg38 coordinates: Expand and add citation, what tool was used.

Line 533-534: Please elaborate and justify

Line 535: Mention the fixed effects used here

Line 546: Please mention the version of the R package used.

Line 564: 'PP' Posterior probability? It would be helpful to redefine this abbreviation here.

Line 568: "Various resources" Implies multiple resources used but only a single citation is provided. Which resources are used?

Line 569-570: First you list onto h38 from h19 then you go back to h19. Justification is needed to understand.

Line 576-577: Is this a supported assumption? It's not immediately obvious why blood is relevant here

Line 587-588: What about regulatory elements

Line 594: Is this just for the 10 loci that were significant in the overall analysis? Were there any sites where it was significant in sex but not overall?

Line 598-599: Here it is controlled for multiple testing - why not in the GWAS? Please elaborate

Line 620: "HREs" It's been a while since these are defined - would make for slightly easier reading to state what this means again.

Line 628: Earlier 1000 Genomes are used for LD - why is a different dataset used here?

Line 631: multiple testing corrections are required here too

Line 643: Is this suggestive significance also corrected for multiple testing, please mention

Line 651: This looks like it comes from the UK biobank. Perhaps consider mentioning it as a limitation in the discussion

Line 656: "horizontal pleiotropy" It is unclear as to what this means - some explanation required

Line 660: "Main analysis" Is this the same as the overall analysis mentioned right before? Please mention

Line 676: Repeating comment as to why no correction for GWAS was performed

Line 691-692: Please elaborate on how was this conducted, to enable the reproduction of analysis

Line 709: Please mention what the model contained

Line 718: Can the ARIC data be accessed too?

(Remarks on code availability)

Reviewer #2

(Remarks to the Author)

(Remarks on code availability)

Reviewer #3

(Remarks to the Author)

In this study, Forster et al study performed a GWAS meta-analysis of 12 specific odours in 21,495 individuals of European ancestry, identifying 10 loci, including seven novel ones. Sex-stratified analyses revealed female-specific and sex-differential loci, with plausible candidate genes linked to olfactory receptors, GPCR signalling, and neural mechanisms. Mendelian randomization analyses highlighted a causal role of AD in impairing odour identification and of bioavailable testosterone on pineapple identification in males. Both results are potentially intriguing due to the well-documented sex differences in odour perception and the established link between olfactory loss and certain neurological diseases. While the analyses are well conducted and the results thoughtfully interpreted, I believe that in some sections the criteria of statistical significance should be revisited or further justified.

Comments:

Ideally, correction for multiple testing should account for the analysis of multiple traits and settings. The same consideration applies to the MR analyses, where FDR was applied separately to each run.

In the quality control step, a $I^2 < 85\%$ represents a relatively high level of heterogeneity and necessitates careful interpretation of results, as well as efforts to investigate and explain potential sources of this variability. Beyond individual SNPs exhibiting heterogeneity, are there any traits characterised by a large number of SNPs with heterogeneity? Could this heterogeneity be predominantly attributed to a specific population sample?

In this context, it would be valuable to present the total odour identification performance across the different samples, with results also stratified by sex.

When claiming replication, an r^2 of 0.3 is generally considered insufficient. Beyond the index variant, do the other associated SNPs within each locus exhibit stronger r^2 values with previously reported associations?

Minor: "(pSIA = 0.09)" is not nominally significant

(Remarks on code availability)

The code is well-organized and adequately documented, making it accessible for other users to run.

Reviewer #4

(Remarks to the Author)

Förster et al. present a GWAS study on human olfactory and focus on the sex-specific signals. The trait was measured by a 'Sniffin' sticks test and the authors have collected the largest GWAS study of the individual smell identification (N = 21,495). They further ran MR analysis and identify the potential causal relationship between other relevant traits and diseases.

As a GWAS study, there are several components missing in this paper. First, what is the heritability of the trait(s). It will be straightforward to run a LDSC regression analysis to check this. Second, is there any significant variant that has MAF < 1%?

To harmonise four cohorts it has to take the intersection of the variants but if there are some low-frequency signals, it will be super valuable to discuss. Third, in terms of finding functionally relevant genes, the current study is more like listing the information for both novel and known loci. It is suggested to run colocalisation analysis using larger eQTL summary statistics from both blood and brain tissues. There are other types of omics data, including methylation QTL and chromatin QTL from blood or brain, which will provide more insights into how the genes are relevant to olfactory functions. Those data can be directly download from here: <https://yanglab.westlake.edu.cn/software/smr/#DataResource> Fourth, while MR is a powerful tool to identify potential causal relationship, without certain prior knowledge, it will be useful to explore the relationship between the olfactory traits with a wide range of complex traits/diseases. The authors might consider running bivariate-LDSC to expand the knowledge on the potential correlations.

Below are some minor comments or questions:

Line #126, Is there any explanation why the signal at locus 7 became insignificant after conditioning on the index variant of locus 8?

Line #176, #189, and #212, please show the creditable set % whenever it is mentioned.

#541, how many loci showed significant heterogeneity between studies? Have the authors run a genetic correlation analysis between each pair of studies to make sure most genetic correlation estimates are close to 1?

#550, did the authors use LD clumping in PLINK to do this?

(Remarks on code availability)

The github repo has covered the main analyses and are detailedly recorded. We noticed that the authored stated "the in-house annotation pipeline will not be provided". While we understand the privacy of the raw data, would that be possible to list some key analyses during these steps?

Reviewer #5

(Remarks to the Author)

(Remarks on code availability)

Version 1:

Reviewer comments:

Reviewer #1

(Remarks to the Author)

This is the second round of revisions. We focussed on the rebuttals provided in the first round of comments.

We appreciate the authors for going through all the comments and addressing them.

1) The response to comment 5 from reviewer 1:

One would still need to account for multiple testing when considering a small number of traits. The paper cited <https://doi.org/10.1093/ije/dyr178> compares studies that use this 5e-8 threshold, but these studies appear to be each focused on a singular trait, hence this citation might not be relevant to the rebuttal argument.

It is not clear how and why the significance threshold is so heavily based on how 'commonly studied' a phenotype is. Further, if the analysis is explanatory then it should be phrased as such. Currently, the abstract does not read that way which can be misleading.

2) response to Line 524

Even though the participants may not have reported relatedness, one wouldn't know unless one accounts (tests or controls) for it. Since the authors state in the rebuttal that ancestry is self-reported, it might be worthwhile to have a quick check.

3) please clarify further what differences in cohorts haven't been accounted for

(Remarks on code availability)

Reviewer #2

(Remarks to the Author)

(Remarks on code availability)

Reviewer #3

(Remarks to the Author)

I appreciate the authors' thorough responses to my comments and the revisions made to the manuscript. My concerns have been adequately addressed, and I am generally satisfied with the improvements. I only have two minor comments:

1. A reference to Table 1 is missing in the main text.
2. As in Table 1, please also include information on which loci pass the Bonferroni-corrected significance threshold (adjusted for the number of analyzed traits) in the tables and figures of the supplementary material (e.g., Supplementary Figures S3, S4).

(Remarks on code availability)

I checked the code in the previous round of revisions

Reviewer #4

(Remarks to the Author)

The authors have addressed my and other reviewers' comments and the manuscript has been substantially improved.

The only remaining concern is that none of diseases tested showed a nominally significant genetic correlation with the odour identification score. The authors proposed the negative results could be due to small heritability estimates of the identification score. I was wondering if the authors have tested other traits with higher heritability such as pineapple identification in females ($h^2 = 0.12$)?

Also, for coffee identification in females ($h^2 = 0.09$), has it been tested against coffee or tea consumption in UKB? These data should be available in https://yanglab.westlake.edu.cn/pub_data.html

(Remarks on code availability)

The analysis and visualization code was updated and well organised.

Reviewer #5

(Remarks to the Author)

(Remarks on code availability)

I share the same view as the primary reviewer.

Version 2:

Reviewer comments:

Reviewer #2

(Remarks to the Author)

From my side, I am contented with this version, and believe it is substantially clearer than the first iteration.

(Remarks on code availability)

Reviewer #4

(Remarks to the Author)

The authors have addressed all my comments and concerns. I don't have further comments.

(Remarks on code availability)

Reviewer #5

(Remarks to the Author)

(Remarks on code availability)

Point by point response to reviewer comments

Reviewer #1

This study investigated the genetics of human olfactory perception through a comprehensive genome-wide association meta-analysis involving over 21,000 individuals of European ancestry. By utilizing a standardized smell test called the 'Sniffin' Sticks,' researchers examined the ability to identify 12 different odours. They identified ten genetic regions linked to specific scents, with seven being newly discovered. Unique to this work, the authors conducted analyses based on sex, finding loci that were either female-specific or showed stronger genetic effects in males. Many of the implicated genes were related to olfactory receptors or pathways involved in cell signalling and neural function. Additionally, the study used Mendelian Randomization techniques to reveal a potential causal link between bioavailable testosterone and the identification of a specific scent (pineapple) in men, as well as a connection between Alzheimer's disease and general olfactory abilities.

The findings provide insight into how genetics may shape differences in smell perception between men and women and suggest a genetic basis for the observed sexual dimorphism in olfaction. The study also investigates connections between olfactory function and neurodegenerative diseases such as Alzheimer's and Parkinson's, with implications for early diagnosis and understanding of disease progression. However, the researchers note limitations, such as a relatively narrow range of scents tested and a sample restricted to European ancestry, pointing to the need for broader, more diverse studies to fully capture the genetic landscape of smell perception.

Authors response: We thank the reviewer very much for the positive evaluation and detailed comments that helped with the improvement of the paper. Please find below a point-by-point response to each comment.

Major comments

Comment 1: *The introduction could be expanded by discussing olfactory perception within the broader context of sensory and cognitive functions. Exploring its connections to memory, decision-making, and emotional responses would emphasize the multifaceted importance of this sense and its relevance to human behaviour and health.*

Authors response:

We agree that these are interesting topics to discuss and included them in the introduction.

Changes in the manuscript:

We added the following paragraph to the introduction (Introduction, 1st paragraph):

"Smell is one of the primary senses in humans. Among the human senses, olfaction is unique due to its direct neuronal connection to the amygdala, a structure of the limbic system taking part in emotional responses and memory¹. Consequently, olfactory cues are known to activate the amygdala and can evoke autobiographical memories². Odours can further be associated with emotions based on past experiences³. These acquired odour preferences can also influence decision making, as avoidance of negatively associated odours is typical⁴."

Comment 2: *A brief mention of other health conditions linked to olfactory dysfunction, such as metabolic disorders, cancer, or infectious diseases, could widen the paper's appeal and emphasize the broad significance of olfactory research.*

Authors response:

We agree and added the primary etiological causes of olfactory dysfunction to the introduction to better reflect the links to other health conditions.

Changes in the manuscript:

We added the following statement (Introduction, 2nd paragraph):

"It is permanently or transiently caused by sinonasal disease, viral infections of the upper respiratory tract, traumatic brain injury of varying degrees, as well as neurodegenerative diseases⁶."

Comment 3: *The reliance on the 'Sniffin' Sticks' test for the measurement of olfactory traits, might limit the breadth of olfactory traits examined. It might be worth discussing how this choice may have affected the scope of the study. Also, consider discussing why these particular smells were chosen.*

Authors response:

Indeed, the comparatively small range of odours tested with the 'Sniffin' Sticks' test is a limitation of our study, which is due to usage of a pre-designed product. We expanded the corresponding paragraph in the discussion to better reflect the selection criteria for odours and the consequences on our findings.

Changes in the manuscript:

We expanded the following paragraph (Discussion, 4th paragraph):

"It needs to be acknowledged that the available 'Sniffin' Sticks' test only represents a very limited excerpt of the detectable odour spectrum, which has been estimated to be larger than 1.72×10^{12} distinguishable olfactory stimuli⁵¹. As the 'Sniffin' Sticks' test is primarily designed for a clinical application, the utilized smells are primarily chosen with regard to cultural familiarity to European individuals, similar intensity and identifiability as well as a bias towards pleasant smells⁵². Therefore, the test is not a representative selection of perceptible olfactory stimuli, as is also demonstrated in the over-representation of nutrition-related odours. The overlap of stick components is low, with dipentene being the only known shared major constituent of two odours, namely orange and lemon (see Supplementary Table S12). Thus, genetic effects of genes more generally involved in odour perception, like *OMP*, *GAP43*⁵³ or *ADCY3*, may have been undetected by our study and potentially require larger sample sizes and broader coverage of the human odour spectrum. This lack of olfactory coverage might also explain why no independent associations with the identification score were found."

Comment 4: *Provide a discussion on how populations of different ancestry could drastically change the outcome of the study, as some 'smells' might be completely foreign to them. For future studies, how should one select the smells to test differences in olfaction, based on the demography? And how one might be able to universalise these effects.*

Authors response:

We expanded the corresponding paragraph in the discussion to explain the challenges when extending our approach to populations with non-European ancestry.

Changes in the manuscript:

The following section was added to the 9th paragraph of the discussion:

"Odour perception can be influenced by cultural background^{63,64}, perhaps a result of differing diets. It would be interesting whether this extends to genetic differences. Investigating the genetic basis of odour identification in other ethnicities is possible by adapting the 'Sniffin' Sticks' test, as has already been done for a variety of populations, usually by changing the descriptors on the answering sheets rather than the chemical composition of the presented odours⁶⁵. However for trans-ethnic meta-analyses, the usage of a test independent of cultural exposure, such as odour resolution is proposed⁶⁶."

Comment 5: *In the methods, the association analysis uses the genome-wide threshold of 5e-8. As a separate test is effectively conducted for each of the 13 odour loci, this significance threshold should be adjusted to account for multiple testing. This may alter the number of significant (novel) loci identified (depending on the correction method used). For example, a rough statistical bonferroni adjustment would shift the significance threshold such that 2 of the novel loci would no longer reach genome-wide significance.*

Authors response:

We thank the reviewer for being attentive to the statistical rigorousness. While we are aware that the adjustment for multiple testing based on the number of traits is common in proteomic and transcriptomic GWAS that investigate thousands of phenotypes, it is, to our opinion, less common in

GWAS typically considering a low number of traits, i.e. the genome-wide significance threshold of 5×10^{-8} was generally accepted by the community as sufficiently conservative resulting in good replication rates (see <https://doi.org/10.1093/ije/dyr178>). Moreover, secondary analyses are often dedicated to assign functionally plausible mechanisms to genetic associations, which - as in our case - increases the credibility of the identified regions. To increase transparency about the statistical reliability, we highlighted SNPs in Table 1 that fulfil the stricter Bonferroni adjusted threshold of $p < 3.85 \times 10^{-9}$, resulting from additional adjustment on the 13 investigated traits. We further included this threshold in Figure 1 and added information about loci passing the stricter threshold to the manuscript text, enabling the reader to make a self-informed decision on the reliability of our results.

We still include loci that do not pass this tightened threshold to increase comparability, as previously performed GWAS of olfaction did also not adjust for the number of traits (see <https://doi.org/10.1016/j.cub.2020.09.012>, <https://dx.doi.org/10.1016/j.cub.2013.07.031>) and the GWAS catalogue also uses the general cut-off of 5×10^{-8} . We would also like to argue that our study is an exploratory analysis of a rarely investigated phenotype. Therefore, all possible candidate loci might be of interest to better understand its underlying genetic mechanisms, in particular since we could assign biologically plausible candidate genes for the majority of our loci.

Changes in the manuscript:

We added the following statement (Results – Genome-wide association meta-analysis, 3rd paragraph):
“Seven of these loci (loci 2, 4, 5, 6, 8, 9, 11) remain significant, when applying a stricter threshold of $p < 3.85 \times 10^{-9}$, accounting for testing 13 odour traits.”

We adapted the methodology of the locus definition (Methods – Locus definition):

“We also considered a stricter threshold of $p < 3.85 \times 10^{-9}$, accounting for testing 13 odour traits.”

We highlighted variants passing the Bonferroni-adjusted threshold in Table 1 and changed Figure 1 to also include this adjusted threshold.

Comment 6: *Since six loci were found to be associated with pineapple and cinnamon, it would be useful to provide more information on the specificity of the associations. Discuss on why these particular odours might be more strongly linked genetically.*

Authors response:

We thank the reviewer very much for this suggestion. When we investigated possible reasons for this phenomenon, we found that heritability estimates of cinnamon ($h^2_{\text{all}} = 0.03$) and pineapple ($h^2_{\text{all}} = 0.07$) were not higher than for other odours. Therefore, we cannot assume a stronger genetical basis of these odours. The overrepresentation of these phenotype associations might therefore be a result of their increased power due to the higher misidentification rates observed for these traits, which is also supported by our power analysis (see Supplementary Figure S17).

Changes in the manuscript:

We incorporated these findings into the discussion (Discussion, 3rd paragraph):

“Six of the ten identified loci were associated with only two odour traits, namely pineapple and cinnamon. These two odours also involve the strongest associations observed in our study and are, together with lemon, the odours showing the lowest identification performances in olfactory studies¹⁴, increasing the power for detecting genetic associations (see Supplementary Figures S17 and S18). Since observed odds ratios of index variants of pineapple and cinnamon were not larger than for other odours, the lower p-values of these associations are a result of the higher minor allele frequencies (MAFs) at these loci.”

Comment 7: *Providing more context about the biological relevance and magnitude of effect sizes (e.g., how meaningful these effects are in practical terms) would be helpful. This is particularly relevant for loci showing strong associations with specific odours like pineapple or cinnamon.*

Authors response:

We agree to improve the interpretability of our results. For this purpose, we calculated effect-direction harmonized odds ratios, that represent the increase in odds to identify an odour correctly per copy of the risk allele. It turned out that several odds ratios are larger than 1.3, which is relatively large

compared to typical single SNP effects of complex diseases. We incorporated these values into the description of the identified loci (see section Results - Single locus results) and added an overview of these effects as Supplementary Table S7.

Changes in the manuscript:

The following sentences were added or modified in the locus description (Results – Single locus results):

Locus 1: “The odds ratio was 1.71 (reference allele: A, 95% confidence interval (CI): 1.41-2.08), meaning each copy of the A allele increases the odds to identify fish correctly by this factor.”

Locus 2: “This index variant exhibited the second strongest effect, with an odds ratio of 2.88 in females (reference allele: G, 95% CI: 2.02-4.08).”

Locus 3: “The variant showed the lowest effect of all index variants with an odds ratio of 1.17 (reference allele: T, 95% CI: 1.11-1.24).”

Locus 4: “The index variant of locus 4 (6p22.1), rs3117345, showed its strongest association in the overall analysis of cinnamon identification ($p = 8.14 \times 10^{-24}$) without sex interaction ($q_{SIA} = 0.74$) and an odds ratio of 1.30 (reference allele: A, 95% CI: 1.23-1.36).”

Locus 5: “Accordingly, this variant showed the strongest effect out of all index variants with the odds ratio reaching 3.76 (reference allele: A, 95% CI: 2.76-5.12).”

Locus 6: “The odds ratio was 1.38 (reference allele: T, 95% CI: 1.31-1.45).”

Locus 8: “The odds ratio was 1.35 (reference allele: A, 95% CI: 1.28-1.41).”

Locus 9: “Locus 9 (11q12.1, index variant: rs61902559) showed the strongest association for pineapple in the overall analysis and also reached genome-wide significance in males ($p_{all} = 2.74 \times 10^{-17}$, $p_{male} = 8.44 \times 10^{-16}$, $p_{female} = 9.98 \times 10^{-6}$), with an odds ratio of 1.24 (reference allele: T, 95% CI: 1.18-1.30).”

Locus 10: “The odds ratio was 1.20 (reference allele: A, 95% CI: 1.13-1.29).”

Locus 11: “Locus 11 (14q11.2, rs2318888) was most strongly associated in the overall analysis of pineapple identification ($p = 1.06 \times 10^{-12}$) here reaching an odds ratio of 1.20 (reference allele: T, 95% CI: 1.14-1.26).”

We also revised the respective methods section accordingly (Methods – Locus definition):

“To derive an intuitive measure of effect, we translated the beta-coefficients of our logistic regression analyses into odds ratios using the allele representing better perception performance as reference.”

Comment 8: *The mechanistic claims between the genetic loci, sex hormones, and olfaction are speculative. It would benefit by discussing past literature if available on functional assays of these mechanistic elements and how they relate to sex-specific olfactory responses. Providing some future directions on how such functional assays could be performed to understand these effects would be beneficial to the wide readership.*

Authors response:

We agree to improve the discussion of this issue. In detail, we expanded our discussion by giving an outlook on potential experiments to verify the proposed mechanisms.

Changes in the manuscript:

We added the following section (Discussion, 6th paragraph):

“In the past, cell culture based assays have been established as a useful tool to identify odorant-receptor interactions^{33,34,45,46}. To investigate our proposed mechanisms, these functional assays could be expanded by applying different levels of sex hormones and observe receptor activity or expression level. To investigate interactions further downstream in the olfactory signal processing pathway, such as the role of *ADCY2* or *GSX2*, it is likely that *in vivo* models will be required.”

Comment 9: *Since the authors did not find strong evidence for neurodegenerative diseases it might be worth reframing the introduction in a way that does not emphasize too heavily on AD and PD.*

Authors response:

We agree that the introduction might be too heavily focussed on neurodegenerative diseases and therefore shortened the corresponding paragraph.

Changes in the manuscript:

We shortened the section introducing the connection to neurodegenerative diseases (Introduction, 2nd paragraph):

"In patients of Alzheimer's disease (AD) or Parkinson's disease (PD) the prevalence of olfactory impairment is greater than 80%^{7,8}. In PD, olfactory loss precedes the onset of the motor symptoms by several years⁹ and is therefore discussed as a potential early marker¹⁰⁻¹²."

Comment 10: *Mention and discuss how environmental factors such as diet, exposure to pollutants, and cultural habits can interact with genetic predispositions to influence olfactory function.*

Authors response:

We agree that a number of environmental factors could affect these traits and that gene-environment interactions are conceivable. Since these mentioned cofactors are notoriously difficult to assess, a comprehensive investigation of their impact could not be performed. Thus, we discussed these potential influencing factors and mentioned potential gene-environment interactions in the limitation section of the revised manuscript.

Changes in the manuscript:

We added the following statement (Discussion, 9th paragraph):

"Further, we were not able to investigate gene-environment interactions. While chronic exposure to a variety of toxins has been reported to be associated with olfactory dysfunction, including heavy metals, pesticides, herbicides, solvents, or chemotherapeutic agents, there is a lack of adequate epidemiological data to perform such genetic interaction analyses⁶."

Comment 11: *It would be worthwhile to provide an evolutionary perspective on why olfactory traits might exhibit strong sexual dimorphisms and how this might relate to survival, reproduction, and social interactions in humans can enrich the narrative and draw connections to evolutionary biology.*

Authors response:

We thank the reviewer for this interesting idea. We expanded our discussion accordingly.

Changes in the manuscript:

The following section was added to the discussion (Discussion, 6th paragraph):

"However, cross-species translations between mice and humans are not straight-forward due to strong species-specific reliance on the sense of smell. This resulted in divergent evolutionary trajectories, as is demonstrated by the higher pseudogenisation rate of ORs and the loss of a functional vomeronasal organ in humans²⁰. This complicates an evolutionary interpretation of the identified sex differences. Sex-differential olfactory capabilities in mice are often directly linked to mate selection and reproductive success⁶⁰. In contrast, the role of olfactory cues in human mate choice and social relations is a topic of ongoing discussion."

Comment 12: *The identification of ten independent loci, including seven novel ones, is an important finding, the results could benefit from more in-depth explanations about why these loci may have been missed in previous studies. This context would help clarify the novelty and importance of the findings.*

Authors response:

Previous analyses often investigated isolated chemicals or the identification score rather than the individual odours of the 'Sniffin' Sticks' test. We adapted the introduction to emphasise this point. We also performed additional power calculations (Supplementary Figure S18) to demonstrate that our study has improved power compared to previous studies, especially when MAF is low or effects are small. As have been seen for other complex traits, we would expect identification of further genetic markers if even larger sample sizes become available in the future.

Changes in the manuscript:

The following statement was added to the introduction (Introduction, 5th paragraph):

"These associations were primarily found for isolated chemicals or overall olfaction rather than individual odours."

The methodology of the power analysis (Methods – Power analysis) was adapted:

"Power analysis to evaluate the impact of different 'Sniffin' Stick' wise case rates and study sample sizes was performed with the function 'genpwr.calc' from the R library 'genpwr'."

Comment 13: *The causal relationship between bioavailable testosterone and pineapple odour identification in males needs context on why this specific relationship might exist is needed. Are there known links between testosterone and olfactory pathways that make this result plausible?*

Authors response:

As recommended by reviewer #3, we adapted the multiple testing correction for the MR analyses. We now apply a two-step FDR approach by adjusting on the number of exposures per outcome in the first step and on the number of outcomes per exposure in the second step (see section Methods - Mendelian randomization analysis). Since our MR effect of testosterone on pineapple did not withstand this more conservative multiple testing correction, we removed this finding in the revised version of the manuscript (see section Results - Two-sample Mendelian randomization).

Minor comments

Line 56-62: *This paragraph would be better placed as the first and then one introduces the neurodegenerative diseases followed by the genes paragraph..this would flow better with the paragraph starting line 70.*

Authors response:

We rewrote this section due to this and other reasons hoping that the flow has been improved. We kindly refer to major comments 1, 2 and 9 for details.

Line 57: *What does "lower detection threshold" mean*

Authors response: Women are able to detect lower concentrations of odours than men. We changed the wording to improve clarity.

Changes in the manuscript:

We rephrased the following statement: "Generally, women exhibit better olfactory identification, better odour discrimination and can detect odours at lower concentrations^{13,14}."

Line 57-58: *It would be better to elucidate a couple of these hypotheses, it reads a bit dry as it is framed now*

Authors response:

We are thankful for the interest and added a short description of some hypotheses. We also pointed out a meta-analysis showing a detailed overview.

Changes in the manuscript:

We expanded the following section: "Several hypotheses have been proposed to explain this phenomenon (for an overview see Sorokowski *et al.*)^{13,15,16}. These include differences caused by gender, for example a higher familiarity of women to culinary odours due to their traditional role in food preparation and biological explanations, like changes in odour sensitivity being caused by fluctuating sex hormone levels during the menstrual cycle."

Line 60-62: *This sentence reads truncated. Consider expanding and rephrasing or making two separate sentences.*

Changes in the manuscript:

We restructured the following statement: "Several hormones serve as transcription regulators¹⁷⁻¹⁹. Therefore, gene-by-sex interactions may play a role in the observed sexual dimorphisms, an issue not investigated so far."

Line 80: *Reframe, as in the current study the sex-stratified analysis actually has fewer number of individuals. Suggesting the current analysis is also limited as this ref 19 one.*

Authors response:

Indeed, sample sizes of sex-stratified analyses are reduced. Our sample size for the sex stratified analyses is comparable to the overall analysis of Gisladdottir *et al.* Accordingly, most of our loci are identified by the overall analysis rather than sex-stratified analysis. To illustrate this issue, we added power calculations for different sample sizes in Supplementary Figure S18.

Changes in the manuscript:

We extended our limitations (Discussion, 9th paragraph): “Due to lower power of the sex-stratified analysis (see Supplementary Figure S18), most of our loci were identified in the overall analysis. Our observation of several suggestive variants with plausible candidate genes indicates that larger genetic meta-analyses will be required to further advance our understanding of the genetic background of olfaction and its sex dimorphisms.”

Line 81: *It would be good to mention the demographics of the populations used in these studies and how the current study in the following paragraph strengthens the existing literature*

Authors response:

We added the ethnicities considered in previous studies. We also mentioned the overlap of our study with the existing literature in the following paragraph.

Changes in the manuscript:

We expanded the following statement: “To date, more than 30 variants with genome-wide significant association with different olfactory traits have been reported^{22–25,27} for European American, African American, Central European and Icelandic populations.”

We added the following statement (Introduction, 6th paragraph): “A partial overlap with previously reported variants was observed.”

Line 90: *This is a big claim. The study hasn't really found extensive evidence to support this claim. Consider rephrasing*

Changes in the manuscript:

We toned down our statement as proposed: “Additionally, we conducted sex-stratified analyses to investigate single nucleotide polymorphism (SNP)-by-sex interactions and their biological background, uncovering a potential mechanism that could contribute to sex dimorphisms of olfaction.”

Line 92-93: *Expand on the bidirectional relationship tested. This phrasing makes it sound like the study only addressed the relationship between odour on disease and not vice-versa (wherein the novel loci were found)*

Changes in the manuscript:

We agree that rephrasing is needed. We modified this section to better represent our bidirectional approach: “Using identified genetic associations, we performed two-sample Mendelian randomization (MR) analyses to test for causal effects of sex hormones on odour identification, and to explore bidirectional causal relationships between olfaction and two major neurodegenerative diseases that are preceded by olfactory impairment, namely AD and PD.”

Line 101-103: *Explain the discrepancy in the sample sizes for the two analyses*

Authors response:

In ARIC, single odour items were only reported for a subset of subjects.

Changes in the manuscript:

We added this explanation to the first paragraph of the results section: “The total sample size was 21,495 (9,909 males, 11,586 females) in the identification score analysis, and 18,895 (8,757 males, 10,138 females) in the twelve single odour analyses. The reduced sample size for the latter analysis is caused by incomplete recording of single odour identifications in the ARIC study.”

Line 210-211: Consider expanding on this result in the discussion, as it is a one-off comment at this stage

Changes in the manuscript:

We expanded the description of this locus (Results – Single locus results, Locus 8): “However, we did not observe associations with coffee identification, indicating that variants reported for coffee consumption might play a stronger role in taste than in smell perception.”

Line 240-241: Might be interesting to link this information with cinnamon specificity in females, in the discussion. See <https://pubmed.ncbi.nlm.nih.gov/23867208/> for more details

Authors response:

We thank the reviewer very much for providing this reference and incorporated it into the description of the locus.

Changes in the manuscript:

We added the following statement (Results – Single locus results, Locus 10): “Chemical components of cinnamon were shown to exhibit a blood pressure lowering effect after ingestion in diabetes patients⁴², but this effect is transmitted through receptors without connection to odour perception.”

Line 352-354: Discuss the potential mechanisms (e.g., impact on olfactory bulbs or neurodegeneration). Further, clarifying why no similar effect was found for Parkinson’s disease may help explain potential differences in disease pathology.

Authors response:

We expanded the discussion of potential mechanisms for the observed effect of AD on the score by explaining the role of *TOMM40*, the driver of this effect. Since we revised our multiple testing strategy, we now also found a significant effect for PD in our main analysis, which, however, was not robust when considering alternative MR methods. Thus, we cannot exclude that the effect is also present for PD.

Changes in the manuscript:

We added the statement to the discussion (Discussion, 8th paragraph): “The associations of AD on odour identification were driven by a variant in the *TOMM40* gene. *TOMM40* codes for a translocase that participates in the import of proteins into mitochondria. Cell models have shown that *TOMM40* induces mitochondrial dysfunction, which can result in neurotoxicity⁶¹. Therefore, we hypothesize that neurodegeneration caused by mitochondrial dysfunction triggers both, olfactory dysfunction and AD development⁶². In our primary analysis, we further observed a causal effect of PD on the identification score in the overall analysis and of AD on the score in females, but these effects were not robust regarding alternative MR methods requiring replication in future studies.”

Line 358-359: It would be helpful to focus on/remind the reader which ones were novel

Changes in the manuscript:

We added the novel loci here as requested: “We identified ten independent genome-wide significant loci of which seven (loci 1, 2, 3, 4, 8, 10, 11) were not previously reported for olfactory traits.”

Line 373-376: Speculative claim. Consider providing evidence for this. In addition, it would be good to provide information in the supplementary on the identification rate per smell

Authors response:

We added additional evidence based on LIFE-Adult data, where the association with the identification score vanishes when pineapple odour is excluded from the score calculation. The number of correct identifications and misidentifications per study is now provided as Supplementary Table S2.

Changes in the manuscript:

Additional evidence for the dependence of the score on the pineapple association was incorporated: “However, we conclude that the score association is driven by the strong pineapple association since the index variant did not even show nominal significance ($p < 0.05$) for odour traits other than pineapple

and in an analysis based on the LIFE-Adult study, the score association vanishes if pineapple is excluded from the score calculation (with pineapple: $p = 5.88 \times 10^{-5}$, without pineapple: $p = 0.06$).

Line 378-382: *Were these odours chosen due to their strong genetic effects, or do they reflect biases in test composition? Addressing this would help readers understand potential limitations in trait selection.*

Authors response:

The 'Sniffin' Sticks' test was primarily designed for clinical application and, to our knowledge, genetic effects were not considered in the odour selection process. We added the criteria for odour selection as well as potential limitations to the manuscript. We like to refer to our answer on major comment 3 for details.

Line 387: *Consider mentioning "only known" as the composition mentioned previously seems to not have the complete information*

Changes in the manuscript:

The suggested change was incorporated: "The overlap of stick components is low, with dipentene being the only known shared major constituent of two odours, namely orange and lemon (see Supplementary Table S12)."

Line 388: *This needs a reference, what studies and what genes are involved, need to be mentioned.*

Changes in manuscript:

We added examples for genes with a more general involvement in odour perception than olfactory receptors:

"Thus, genetic effects of genes more generally involved in odour perception, like *OMP*, *GAP43*⁵³ or *ADCY3*, may have been undetected by our study and potentially require larger sample sizes and broader coverage of the human odour spectrum."

Line 389-390: *It might be more nuanced than this, consider rephrasing the explanation*

Changes in manuscript:

We rephrased the sentence to clarify that this is only one possible explanation:

"Thus, genetic effects of genes more generally involved in odour perception, like *OMP*, *GAP43*⁵³ or *ADCY3*, may have been undetected by our study and potentially require larger sample sizes and broader coverage of the human odour spectrum."

Line 407-408: *Overstating as only two variants were found*

Changes in manuscript:

We rephrased the sentence to make our statement more specific:

"The observed sex-differential and sex-specific variants suggest that genetic mechanisms might contribute to the sex differences in human olfaction¹³."

Line 428: *Mention that this is only for this context of the study, the smells tested here. It might not be a universal statement*

Changes in manuscript:

We added the requested contextualisation: "To further clarify the possible role of hormones in olfactory performance of the investigated odours, we performed MR analyses. As we did not observe significant causal relationships, the tested sex hormones might not have a general causal impact on olfactory identification."

Line 471: *elaborate on the summary stats*

Changes in manuscript:

We expanded the following sentence by including a list of summary statistics:

"This GWAMA combined summary statistics of additive genetic effects on odour perception (consisting of SNP identifier, genetic position, effect allele, alternative allele, effect allele frequency, sample size,

effect estimate, standard error, p-value and imputation quality) of the following four cohorts: the Rhineland Study, CHRIS⁶⁷ (Cooperative Health Research In South Tyrol), LIFE-Adult⁶⁸ (Leipzig Research Center for Civilization Diseases) and ARIC⁶⁹ (Atherosclerosis Risk in Communities).”

Line 476: *What is the threshold for European ancestry? Bi or uniparental and what is the time cut-off*

Authors response:

In the ARIC study, ethnicity was assigned based on self-reporting.

Changes in manuscript:

We clarified the determination of ethnicity for the ARIC study (Methods – Contributing studies, genotyping and imputation, 1st paragraph): “ARIC contains individuals from the US with self-reported African American and European background, but only the individuals with European ancestry were included in this study due to the small sample size of the African American subgroup.”

Line 494-496: *Is this accounted for in the analysis, as this might inflate results and/or introduce artefacts*

Authors response:

Possible study heterogeneity is accounted for by calculating I² statistics and removing variants with large values.

Line 505-506: *Was the order of the smells the same or not for each individual*

Authors response:

The order of smells was consistent across individuals corresponding to the manufacturer’s recommendation.

Changes in manuscript:

We clarified the consistent order of smells in the methodology (Methods – Measurement of odour identification): “Provided options were the same across all studies and the order of smells was consistent across individuals.”

Line 506-508: *Is there a way to provide information on how the identification of each smell was correlated with the other smells? For example, does identifying pineapple mean one never identifies rose etc? Would be helpful to state this somewhere*

Authors response:

In LIFE-Adult, no strong correlations between the odours were observed. We provided these correlations in Supplementary Figure S1.

Changes in manuscript:

We described our methodology to obtain the pairwise correlation of single odour perceptions (Methods – Measurement of odour identification): “Pearson correlation coefficients of single odour perceptions were determined in LIFE-Adult.”

We mention our results in the manuscript (Results - Genome-wide association meta-analysis, 1st paragraph): “The individual odour perceptions showed no strong correlations (maximal |r| = 0.2, see Supplementary Figure S1).”

Line 517-518: *Mention which pens, and across which studies. Especially relevant for pineapple and cinnamon*

Authors response:

This information is provided in the corresponding Supplementary Table S12. Pineapple and cinnamon were not affected.

Changes in manuscript:

We added the link to Supplementary Table S12: "It should further be noted that compositions of specific pens have changed over time, further hindering the determination of the exact components (see Supplementary Table S12)."

Line 521: *Why not interactive models, as one would lose information otherwise*

Authors response:

Gene-environment interaction studies generally require a larger sample size than studies only focussing solely on the genetic main effect (<https://doi.org/10.1093/aje/kwx227>). As our sample size is limited, we would like to refrain from genetic interaction testing due to lack of power.

Line 524-525: *but no control of genetic relatedness?*

Authors response:

Adjustment for genetic relatedness was performed for the CHRIS study which is characterised by related individuals. For the other studies no correction was necessary, as the participants are not related.

Line 526: *Pairwise genetic relatedness? This seems quite vague.*

Authors response:

Yes, the correction is performed based on the pairwise genetic relatedness.

Changes in manuscript:

We added this specification to the manuscript: "Accordingly, the ARIC study additionally adjusted for study centre, the Rhineland study for the first ten genetic principal components and CHRIS for pairwise genetic relatedness."

Line 531-532: *Quite a bit vague - what does "harmonising" entail? Would not be able to replicate this analysis based on this description. Does it control for the different microarrays/ imputation approaches or how the different studies account (or do not account) for shared ancestry?*

Authors response:

Harmonization consists of the lifting to a common genome build and the application of common quality filters prior to the meta-analysis. As such, the harmonization does not adjust on study-wise differences of microarrays or imputation reference. We changed the wording to better highlight that the following section describes the harmonization procedure.

Changes in manuscript:

We adapted the wording in this paragraph to improve clarity about the contents of our harmonization procedure:

"Prior to meta-analysis, single study association results were harmonized by the following steps. Positions of variants from ARIC and Rhineland were lifted from hg19 to hg38 coordinates with the 'rtracklayer' package (version 1.62.0, function 'liftOver')⁸⁰. For these two studies, variants with an allele frequency deviating more than 20% from the European 1000 Genomes reference were removed, as large deviances indicate errors in genotyping or imputation. Next, monomorphic and duplicated variants were removed for all studies, as well as variants with minor allele count < 6. After harmonization, fixed-effect meta-analysis, i.e. an analysis under the assumption that the true effect is homogeneous across studies, was performed with METAL⁸¹ (version from March 25, 2011) using inverse-variance weighting. Between-study heterogeneity was assessed with I² statistics."

hg19 to hg38 coordinates: *Expand and add citation, what tool was used.*

Changes in manuscript:

Tool and citation were added to the following sentence: "Positions of variants from ARIC and Rhineland were lifted from hg19 to hg38 coordinates with the 'rtracklayer' package (version 1.62.0, function 'liftOver')⁸⁰."

Line 533-534: *Please elaborate and justify*

Authors response:

Strong deviations from established genetic references are often a sign of genotyping or imputation errors rather than an actual difference in the population.

Changes in manuscript:

We added our reasoning to the manuscript: "For these two studies, variants with an allele frequency deviating more than 20% from the European 1000 Genomes reference were removed, as large deviances indicate errors in genotyping or imputation."

Line 535: *Mention the fixed effects used here*

Authors response:

We are uncertain, which information is requested by the reviewer here. We performed fixed-effect meta-analysis and determined I^2 statistics to evaluate heterogeneity. We added the latter fact hoping that this is meant by the reviewer's request. We also specified the underlying assumption of the fixed effect model, i.e., that true effect sizes are homogeneous across studies. Otherwise, we would respectfully ask for clarification of the request.

Changes in manuscript:

We extended the following statement: "After harmonization, fixed-effect meta-analysis, i.e. an analysis under the assumption that the true effect is homogeneous across studies, was performed with METAL⁸¹ (version from March 25, 2011) using inverse-variance weighting. Between-study heterogeneity was assessed with I^2 statistics."

Line 546: *Please mention the version of the R package used.*

Authors response:

Regrettably, the package does not provide versions. We added the date of the last update as a substitute.

Changes in manuscript:

We expanded the following sentence: "The R package 'Miamiplot'⁸³ (version from February 24, 2021) was used for this purpose."

Line 564: *'PP' Posterior probability? It would be helpful to redefine this abbreviation here.*

Changes in manuscript:

We reintroduced the abbreviation: "The posterior probability (PP) for each variant was then calculated by dividing its ABF by the sum of the ABFs of all variants at the locus."

Line 568: *"Various resources" Implies multiple resources used but only a single citation is provided. Which resources are used?*

Authors response:

The citation refers to our analysis pipeline described in a previous publication. The used bioinformatic resources, namely the GWAS catalogue, Ensemble, CADD and GTEx, are cited at the appropriate position in the paragraph.

Line 569-570: *First you list onto h38 from h19 then you go back to h19. Justification is needed to understand.*

Authors response:

As imputation references were different across studies, we needed to lift them on a common genome build. We decided to report our primary results based on hg38 to enable easier reuse of our data by other researchers. The resources utilized by our annotation pipeline are based on hg19. We added the limitation to the methodology to make this point clearer to the reader.

Changes in manuscript:

We rephrased the following sentence: "For compatibility with our annotation procedure and available reference data, we down-lifted genetic positions from hg38 to hg19 coordinates for this purpose."

Line 576-577: *Is this a supported assumption? It's not immediately obvious why blood is relevant here*

Authors response:

Blood is not directly relevant from a biological context but has the highest power in eQTL analyses.

Changes in manuscript:

We adapted the wording to better reflect the selection criteria (Methods – Colocalization analysis, 2nd paragraph): "As tissues, we considered blood, due to its high power, and brain tissue due to its connection to olfactory signal processing and analysed eQTLs^{97,98}, mQTLs^{99,100} and splicing quantitative trait loci (sQTLs)⁹⁸ using data of European ancestry."

Line 587-588: *What about regulatory elements*

Authors response:

Regulatory elements are indeed of high interest and are part of our annotation procedure. As we are considering all genes in a broad genetic region around associated variants, cis-regulated genes are ensured to be considered for manual candidate selection. Further, regulatory effects of variants are covered by our colocalization analysis with eQTL data. While trans-effects are annotated during the bioinformatic annotation step, we did not consider them for candidate gene selection due to their unclear mechanistic impact.

Changes in manuscript:

We rephrased the following sentence to better convey the intended goal of the analysis: "During candidate gene identification, we further tested for overlapping signals between identified odour associations and QTLs for different molecular traits to identify regulatory effects."

Line 594: *Is this just for the 10 loci that were significant in the overall analysis? Were there any sites where it was significant in sex but not overall?*

Authors response:

The analysis was performed for all 10 independent genome-wide significant loci, regardless of the subgroup in which the best association was observed. Loci 2 and 10 only showed an association in females but not in males or overall (see Figure 1 and Table 1), which is concordant with the observed sex specific effects.

Changes in manuscript:

We adapted the wording to improve clarity: "Sex interaction analyses of all index variants of independent loci were performed according to the procedure described by Winkler *et al.*⁹⁴ with the extension for related individuals."

Line 598-599: *Here it is controlled for multiple testing - why not in the GWAS? Please elaborate*

Authors response:

For comparison of sexes, we used the olfactory trait associated at this locus. Thus, multiple testing correction was performed accounting for the number of loci analysed. For the GWAS, we would like to refer to our changes in response to major comment 5.

Line 620: *"HREs" It's been a while since these are defined - would make for slightly easier reading to state what this means again.*

Changes in manuscript:

We agree and reintroduced the abbreviation for the hormone response elements: "At loci with sex-differential or sex-specific index variants, a search of genes with nearby hormone response elements (HREs) was performed. The genes were looked up in published gene lists for oestrogen response elements (EREs)¹⁰³ and androgen response elements (AREs)⁵⁷."

Line 628: *Earlier 1000 Genomes are used for LD - why is a different dataset used here?*

Authors response:

The choice of LIFE-Adult as LD reference for COJO was motivated based on the recommendations from the creators of the COJO method suggesting the usage of a participating cohort rather than the imputation reference.

Changes in manuscript:

We added the reasoning of our choice of the LD reference to the manuscript: "The LIFE-Adult-Study was used as LD reference for this purpose, as it was the largest participating cohort with accessible individual level genotype data."

Line 631: *multiple testing corrections are required here too*

Authors response:

For the analysis of secondary signals, we considered the best associated trait only, because loci were odour specific. Thus, multiple testing correction does not need to account for multi-trait analyses. We therefore used the mentioned threshold. We would like to acknowledge that even with this threshold, no secondary signals were found.

Changes in manuscript:

To improve clarity, we added in the respective methods section: "Conditional analysis was performed for the best associated trait."

Line 643: *Is this suggestive significance also corrected for multiple testing, please mention*

Authors response:

We used a cut-off of 10^{-6} as a threshold for suggestive significance. We intended this as an exploratory analysis to provide an outlook on potential candidates that require verification in larger studies. As we found biologically interesting genes like *ADCY3* which is directly connected to olfactory signal transduction we believe that this analysis is of interest for other researchers. We mention in the methodology that these effects require validation in subsequent studies.

Changes in manuscript:

We clarified the need for further validation in the methodology (Methods – Analysis of suggestive and rare variants): "Respective results are reported as supportive evidence requiring further validation."

Line 651: *This looks like it comes from the UK biobank. Perhaps consider mentioning it as a limitation in the discussion*

Authors response:

Indeed, this study relies on UKBB data. Two-sample Mendelian randomization approaches are recommended and this study provides the largest sample size for these traits as far as we know. The authors are uncertain about the limitation seen by the reviewer and would kindly ask for clarification here.

Line 656: *"horizontal pleiotropy" It is unclear as to what this means - some explanation required*

Authors response:

We thank the reviewer for pointing out this unexplained terminology. Horizontal pleiotropy refers to effects of an instrumental variant that are not mediated by the investigated exposure, which would be a violation of MR assumptions.

Changes in manuscript:

We added this explanation to the manuscript: "To reduce the risk of horizontal pleiotropy, i.e. effects not mediated by the exposure, as best as possible, we restricted the list of instruments to variants in regions of ± 250 kb around genes of the steroid hormone pathway as defined in KEGG (*hsa00140*)¹⁰⁸."

Line 660: "Main analysis" Is this the same as the overall analysis mentioned right before? Please mention

Authors response:

This term refers to our primary Mendelian randomization analysis followed by sensitivity analyses to check the robustness of the findings.

Changes in manuscript:

To improve clarity, we now use the term "primary MR analysis": "In our primary MR analysis, we used the resulting list of strong instruments of the respective analysis groups (male, female, overall)."

Line 676: Repeating comment as to why no correction for GWAS was performed

Authors response:

We kindly refer to our answer for major comment 5 for a detailed explanation of the performed changes.

Line 691-692: Please elaborate on how was this conducted, to enable the reproduction of analysis

Changes in manuscript:

We agree to provide additional information about our harmonisation procedure: "We performed harmonization of effect alleles by changing the sign of the effect for variants with inverted alleles."

Line 709: Please mention what the model contained

Authors response:

Our power analysis was performed under the assumption of correctness for the additive genetic model. No covariate effects were considered.

Changes in manuscript:

We added this information to the manuscript: "Power calculation was performed under the assumption of correctness of the additive genetic model. Covariate effects were not considered."

Line 718: Can the ARIC data be accessed too?

Authors response:

We thank the reviewer for making us aware of the missing data availability statement and added this to the revised draft.

Changes in manuscript:

We added the missing data availability statement for the ARIC study: "ARIC data from visit 1 to visit 5 are available through the Biologic Specimen and Data Repository Information Coordinating Center (BioLINCC). Data that are not yet available through BioLINCC are available upon request through the ARIC Coordinating Center at the University of North Carolina."

Reviewer #2

Authors response:

We sincerely thank the reviewer for participating in the review process and acknowledge the reviewer's contribution to the refinement of the manuscript.

Reviewer #3

In this study, Forster et al study performed a GWAS meta-analysis of 12 specific odours in 21,495 individuals of European ancestry, identifying 10 loci, including seven novel ones. Sex-stratified analyses revealed female-specific and sex-differential loci, with plausible candidate genes linked to olfactory receptors, GPCR signalling, and neural mechanisms. Mendelian randomization analyses highlighted a causal role of AD in impairing odour identification and of bioavailable testosterone on pineapple identification in males. Both results are potentially intriguing due to the well-documented sex differences in odour perception and the established link between olfactory loss and certain neurological diseases. While the analyses are well conducted and the results thoughtfully interpreted, I believe that in some sections the criteria of statistical significance should be revisited or further justified.

Authors response:

We thank the reviewer very much for the positive feedback and the helpful and constructing comments, helping us to improve the manuscript. Our responses to the individual comments can be found below.

Comment 1: *Ideally, correction for multiple testing should account for the analysis of multiple traits and settings. The same consideration applies to the MR analyses, where FDR was applied separately to each run.*

Authors response:

We agree that statistical rigorousness of GWAS results is of high importance. Indeed, in our previous version of the manuscript, we did not additionally correct for multiple trait comparisons, since to our impression, the usual cut-off of genome-wide significance (5×10^{-8}) was considered as sufficiently conservative by the community ensuring high replication rates. As our goal was to present an exploratory investigation of a rarely investigated trait, we believe that loci fulfilling this criterion are of interest to the reader as we often can support respective loci by biologically plausible genes increasing the credibility of the loci. Moreover, previous studies of the olfactory phenotype did also not adjust for the number of traits (see Gisladottir et al. <https://doi.org/10.1016/j.cub.2020.09.012> and McRae et al. <https://dx.doi.org/10.1016/j.cub.2013.07.031>).

To account for the reviewer's wish, we now highlight loci in table 1 that remain significant when performing additional Bonferroni correction for the 13 investigated traits (i.e. $p < 3.85 \times 10^{-9}$). We further modified Figure 1 to display this more stringent threshold. We also emphasize this issue in the manuscript. We decided to not adjust for the number of sex-settings, as these analyses are not independent and a Bonferroni correction would be clearly overconservative here.

Regarding Mendelian randomization analyses, we revised our testing correction to a two-step approach. In the first step, we apply FDR correction for each outcome on the number of exposures tested. In the second step we additionally adjust the corrected p-values from the first step per exposure on the number of outcomes. This correction is applied for each analysis mode and sex-setting separately. For example, when investigating the effects of sex hormones on odour identification, we adjust p-values for each odour for the three (four in men) tested sex hormones. In the second step, these adjusted p-values are then corrected for each hormone on the thirteen investigated olfactory traits. We further tested the robustness of MR findings with the help of other MR methods.

Further, because literature only supports an influence of neurodegenerative diseases on total olfactory ability rather than individual odours, we now limit our investigation of the effects of neurodegenerative diseases on olfaction to the identification score as the only relevant outcome variable.

Changes in manuscript:

We added the more stringent multiple testing correction to the locus definition (Methods – Locus definition):

“We also considered a stricter threshold of $p < 3.85 \times 10^{-9}$, accounting for testing 13 odour traits.”

We mentioned which loci surpass the multiple testing adjusted threshold in the manuscript text (Results – Genome-wide association meta analysis, 3rd paragraph):

“Seven of these loci (loci 2, 4, 5, 6, 8, 9, 11) remain significant, when applying a stricter threshold of $p < 3.85 \times 10^{-9}$, accounting for testing 13 odour traits.”

We also highlighted variants passing the adjusted threshold in Table 1 and added the stricter cut-off to Figure 1.

We revised our methodology of the Mendelian randomization to reflect our new multiple testing approach (Methods – Mendelian randomization analysis, 3rd paragraph):

“Multiple testing correction was performed for each sex-setting separately with a two-step procedure. At the first step, an FDR correction was performed for each odour outcome over the number of hormone exposures using the Benjamini & Hochberg procedure⁹⁵. In the second step, the adjusted p-values from the first step were FDR corrected per exposure on the number of outcomes tested.”

We rewrote the results section for the MR analysis as our previously reported effects of BAT on pineapple was lost under this more stringent multiple testing correction and additional effects for neurodegenerative diseases on olfaction were observed (see section Results - Two sample Mendelian randomization).

We adapted our discussion of the MR to reflect these changes (Discussion, 7th and 8th paragraph):

“To further clarify the possible role of hormones in olfactory performance of the investigated odours, we performed MR analyses. As we did not observe significant causal relationships, the tested sex hormones might not have a general causal impact on olfactory identification. However, due to the high number of postmenopausal females with oestrogen levels below the detection threshold⁴⁹, we were unable to investigate the effects of oestrogen in women, although this could be a potential contributor to their observed better olfaction performance¹³.”

Finally, we found no support for a causal impact of olfactory impairment on neurodegenerative diseases. Conversely, a high genetic risk for AD showed a negative causal effect on overall odour identification. This suggests that although odour misidentification is a prequel of both, AD and PD, it does not constitute a driving factor of these diseases. Instead, the negative effect of AD on odour identification points to an underlying shared mechanism of olfactory impairment and AD. The associations of AD on odour identification were driven by a variant in the *TOMM40* gene. *TOMM40* codes for a translocase that participates in the import of proteins into mitochondria. Cell models have shown that *TOMM40* induces mitochondrial dysfunction, which can result in neurotoxicity⁶¹. Therefore, we hypothesize that neurodegeneration caused by mitochondrial dysfunction triggers both, olfactory dysfunction and AD development⁶². In our primary analysis, we further observed a causal effect of PD on the identification score in the overall analysis and of AD on the score in females, but these effects were not robust regarding alternative MR methods requiring replication in future studies. MR analyses using traits of mild cognitive impairment as a preclinical phenotype could also be of added value here.”

Comment 2: *In the quality control step, a $I^2 < 85\%$ represents a relatively high level of heterogeneity and necessitates careful interpretation of results, as well as efforts to investigate and explain potential sources of this variability. Beyond individual SNPs exhibiting heterogeneity, are there any traits characterised by a large number of SNPs with heterogeneity? Could this heterogeneity be predominantly attributed to a specific population sample?*

Authors response:

We agree to performed additional analyses to address this issue.

First, to identify whether a single study is the driver of heterogeneity, we performed a sensitivity analysis. For this purpose, we considered the ten independent index variants and reperformed the meta-analysis by sequentially leaving out one of the studies. It turned out that heterogeneity was not driven by a single study (see Supplementary Figure S20).

We further analysed the fraction of SNPs with $I^2 \geq 85\%$ in the overall analysis of each trait. We observed the highest fractions in pineapple, the identification score and cinnamon. Lowest fractions of heterogeneous variants were observed for banana, coffee and orange (see Supplementary Figure S19). However, the observed differences were small in size (i.e. percentages ranged between 0.06%-0.08%). We therefore conclude that heterogeneity is similar across traits.

Changes in manuscript:

We performed additional investigations to identify potential drivers of heterogeneity and added them to the methodology (Methods – Analysis of heterogeneity):

“As the ‘Sniffin’ Sticks’ test is not highly standardized, we performed analyses of the heterogeneity. We investigated the fraction of variants with $I^2 \geq 85\%$ for the overall analysis of each trait. QC was applied prior to this analysis but without filtering for heterogeneity. For the index variants of independent loci, we further performed a sensitivity analysis to identify studies driving heterogeneity. For this purpose, we sequentially left out a single study and recalculated the heterogeneity. It turned out that heterogeneity was not caused by a specific study, and the fraction of heterogeneous variants was similar across the investigated traits (see Supplementary Figures S19 and S20).”

Comment 3: *In this context, it would be valuable to present the total odour identification performance across the different samples, with results also stratified by sex.*

Authors response:

We agree that this is a valuable additional information. We added these data as Supplementary Table S2.

Comment 4: *When claiming replication, an r^2 of 0.3 is generally considered insufficient. Beyond the index variant, do the other associated SNPs within each locus exhibit stronger r^2 values with previously reported associations?*

Authors response:

We agree that this threshold is too low to claim replication. Actually, we used this cut-off for novelty analysis, and therefore, deliberately used a low threshold to be conservative with the claim of novelty.

Of note, index variants of already known loci typically showed stronger LD than $r^2 = 0.3$. The index variant of locus 9 shows an r^2 -value 0.76 with the previously reported variant while our index variants of loci 5 and 6 are identical to the reported ones. We added this information to the manuscript to improve clarity.

Finally, we did not observe genome-wide significant variants after conditioning on the literature variants, indicating that other associated variants (if present) are in LD with the previously reported SNPs.

Changes in manuscript:

We added the r^2 value for locus 9 (section Single locus results – Known loci, 4th paragraph): “The index variant is in LD with a variant previously reported for β -ionone sensitivity by McRae *et al.*²³ (rs7943953, $r^2 = 0.76$).”

We changed the wording for loci 5 and 6 (section Single locus results – Known loci, 2nd and 3rd paragraph):

“This variant was previously reported as an index variant by Gisladdottir *et al.* with a genome-wide significant association for fish intensity and a suggestive association for fish naming²⁴.”

“Rs317787 was previously reported as an index variant of a locus associated with cinnamon naming problems by Gisladdottir *et al.*²⁴ and is located within an OR gene cluster.”

We changed the wording of the methodology (Methods - Credible sets and bioinformatic annotation of variants, 2nd paragraph):

“Annotation with the GWAS catalogue was used to assess novelty. Index variants were considered unreported when they were not in LD ($r^2 > 0.3$) with previously reported variants for the respective trait.”

Minor: “($p_{SIA} = 0.09$)” is not nominally significant

Authors response:

We thank the reviewer for pointing this out and regret the lack of clarity here. The p-value of 0.09 is the FDR adjusted p-value while the nominally significant observation was referring to the p-value prior to correction. We now refer to the adjusted p-values as q_{SIA} and use p_{SIA} to indicate p-values prior to FDR correction.

Changes in manuscript:

We added the unadjusted p-value, which is nominally significant: "A nominally significant sex interaction not withstanding multiple testing correction was observed ($p_{\text{SIA}} = 0.03$, $q_{\text{SIA}} = 0.09$)."

Throughout the manuscript, we replaced p_{SIA} with q_{SIA} where FDR corrected p-values are reported.

Reviewer #3 (Remarks on code availability):

The code is well-organized and adequately documented, making it accessible for other users to run.

Authors response: We thank the reviewer very much for code review and the positive feedback.

Reviewer #4

Förster et al. present a GWAS study on human olfactory and focus on the sex-specific signals. The trait was measured by a 'Sniffin' sticks test and the authors have collected the largest GWAS study of the individual smell identification (N = 21,495). They further ran MR analysis and identify the potential causal relationship between other relevant traits and diseases.

Authors response:

We thank the reviewer very much for the helpful and constructive comments and the suggestions to use additional bioinformatic resources. A response to each individual point can be found below.

Comment 1: *As a GWAS study, there are several components missing in this paper. First, what is the heritability of the trait(s). It will be straightforward to run a LDSC regression analysis to check this.*

Authors response:

We agree that this is a valuable additional information. As suggested, we performed LDSC regression and added the heritability estimates as Supplementary Table S4.

Changes in manuscript:

We added the calculation of heritability to the manuscript (Methods - Post-analysis quality control, 2nd paragraph):

"Heritability of each trait and subgroup was estimated with LD Score regression. We used LDSC (v. 1.0.1)⁸² for this purpose. SNPs were included based on the default quality criteria of the tool. The provided LD scores for European populations were used as reference."

We added heritability estimates to the results section in the manuscript (Results – Genome-wide association meta-analysis, 2nd paragraph):

"Highest heritabilities were observed for pineapple identification in females ($h^2 = 0.12$), coffee identification in females ($h^2 = 0.09$) and pineapple identification in males ($h^2 = 0.08$, see Supplementary Table S4)."

Comment 2: *Second, is there any significant variant that has MAF < 1%? To harmonise four cohorts it has to take the intersection of the variants but if there are some low-frequency signals, it will be super valuable to discuss.*

Authors response:

Although we agree that an analysis of rare variants is interesting, these variants are often difficult to analyse with small studies due to the lack of power. Often, rare variants were single SNP associations without support in the regional association analysis. This is due to statistical models becoming instable for low MAF variants, in particular for small sample sizes. We performed additional power analyses for different sample sizes at different MAF values (see Supplementary Figure S18). These analyses indicate insufficient power for MAF < 1% except for strong effects.

However, we agree that rare missense mutations would constitute variants of higher prior probability of association. Thus, we limited our search for rare genome-wide significant variants to known missense mutations. In total, we identified 23 rare variants of genome-wide significance. However, none of these variants constitute missense mutations. Since our study was not designed to address the challenges of rare variant analysis, we have chosen not to report these results to avoid the false impression of validity but mention this analysis in the methods section.

Changes in manuscript: We added the following statement (Methods – Analysis of suggestive and rare variants): "We further searched genome-wide significant associations below MAF < 1% for possible missense mutations but no such mutations were found."

Comment 3: *Third, in terms of finding functionally relevant genes, the current study is more like listing the information for both novel and known loci. It is suggested to run colocalisation analysis using larger eQTL summary statistics from both blood and brain tissues. There are other types of omics data, including*

methylation QTL and chromatin QTL from blood or brain, which will provide more insights into how the genes are relevant to olfactory functions. Those data can be directly download from here: <https://yanglab.westlake.edu.cn/software/smr/#DataResource>

Authors response:

We thank the reviewer much for this suggestion. We performed colocalization analyses with these data. Results were added as Supplementary Table S10.

Changes in manuscript:

We added this colocalization analysis to our methodology (Methods – Colocalization analysis, 2nd paragraph):

“During candidate gene identification, we further tested for overlapping signals between identified odour associations and QTLs for different molecular traits to identify regulatory effects. As tissues, we considered blood, due to its high power, and brain tissue due to its connection to olfactory signal processing and analysed eQTLs^{97,98}, mQTLs^{99,100} and splicing quantitative trait loci (sQTLs)⁹⁸ using data of European ancestry. QTL summary statistics for the ten independent loci with olfactory association were extracted with SMR (v1.3.1)¹⁰¹ and lifted to hg38 coordinates with VCF-liftover¹⁰². A test for colocalization was performed if at least 50 variants intersected between olfactory and QTL summary statistics. Colocalization between odour identification and QTLs was assumed when $PP(H4) > 80\%$. We primarily reported identified colocalizations for candidate genes of the respective loci and genes with known olfactory function.”

We mention colocalizations with eQTL or mQTLs for genes with olfactory connection in the section ‘Single locus results’:

Locus 4: “A colocalization with a methylation quantitative trait locus (mQTL) annotated with *OR2J3* in brain was observed (see Supplementary Table S10).”

Locus 6: “The signal for cinnamon identification in males colocalized with mQTLs in blood annotated with *OR51I2* and *OR52B4* (see Supplementary Table S10).”

Locus 8: “Colocalizations of our pineapple signal and mQTLs annotated with *OR4A16*, *OR4C16* and *OR8U8* in brain were observed.”

Locus 9: “The signal for pineapple identification in the overall analysis colocalized with blood mQTLs annotated with *OR10V1*, *OR4D6*, *OR5A2* and *OR5B12* and brain mQTLs annotated with *OR5A1* and *OR5A2* (see Supplementary Table S10).”

Comment 4: *Fourth, while MR is a powerful tool to identify potential causal relationship, without certain prior knowledge, it will be useful to explore the relationship between the olfactory traits with a wide range of complex traits/diseases. The authors might consider running bivariate-LDSC to expand the knowledge on the potential correlations.*

Authors response:

We agree that this analysis could provide useful insights into the connection of olfactory impairment with other diseases. We performed genetic correlation analyses between the olfactory score (sex-combined analysis) and common complex diseases, as the overall ability to smell is more likely to be associated with diseases than single odours. We tested obesity, hypertension, coronary atherosclerosis, osteoporosis, stroke, depression, inflammatory bowel disease, multiple sclerosis, diabetes (type1 and type2), asthma, osteoarthritis, rheumatoid arthritis, PD and AD. None of these diseases showed a nominally significant genetic correlation with the odour identification score, probably due to small heritability estimates of the identification score. Results are provided in the new Supplementary Table S23. We will be happy to extend the list of considered diseases if it is deemed insufficient or if the reviewer would like to see a specific hypothesis tested.

Changes in manuscript:

We added our methodology for the genetic correlation analysis of the olfactory score with complex diseases (Methods – Genetic correlation with complex diseases):

“We screened for genetic correlations between the sex-combined associations of the odour identification score and other common complex diseases. Respective summary statistics were retrieved for AD, asthma, coronary atherosclerosis, depression, hypertension, inflammatory bowel disease, multiple sclerosis, obesity, osteoarthritis, osteoporosis, PD, rheumatoid arthritis, stroke, type 1

diabetes and type 2 diabetes from the pan-UKBB website¹⁰⁵. Genetic correlations were estimated with LDSC (v. 1.0.1)^{82,106} by using the provided LD score reference for European populations.”

We added the results of this analysis (Results – Two-sample Mendelian randomization, 6th paragraph): “We searched for additional phenotypes with possible connection to olfaction by analysing genetic correlations between the sex-combined associations of the identification score and complex diseases. None of the tested diseases showed nominally significant correlations (see Supplementary Table S23).”

minor comments or questions

Line #126, *Is there any explanation why the signal at locus 7 became insignificant after conditioning on the index variant of locus 8?*

Authors response:

Index variants of loci 7 and 8 are in weak LD ($r^2 = 0.12$) and the observed association at locus 7 is likely driven by this linkage disequilibrium. Consequently, the association is no longer observable when conditioning on the effect of locus 8.

Changes in manuscript: We rewrote the sentence to clarify this connection: “In contrast, conditioning on the index variant of locus 8 results in a loss of genome-wide significance for locus 7 ($p_{\text{cond}} = 0.63$) probably due to mild LD between the variants ($r^2 = 0.12$).”

Line #176, #189, and #212, *please show the credible set % whenever it is mentioned.*

Changes in the manuscript:

We added the percentages to the credible sets as suggested.

#541, *how many loci showed significant heterogeneity between studies? Have the authors run a genetic correlation analysis between each pair of studies to make sure most genetic correlation estimates are close to 1?*

Authors response:

The index variants of loci 5, 8 and 11 showed nominally significant heterogeneity.

While pairwise genetic correlation of single study summary statistics would indeed be an important tool to ensure quality, we were unable to perform this as the mean X^2 values for single studies summary statistics were below the recommended threshold for bivariate LDSC.

Changes in manuscript:

We now mention loci with significant heterogeneity (Results – Genome-wide association analysis, 3rd paragraph):

“Nominally significant heterogeneity was observed for index variants of loci 5, 8 and 11.”

#550, *did the authors use LD clumping in PLINK to do this?*

Authors response:

In our analysis, physical locus definition was performed to avoid overlapping signals, i.e., no LD clumping was performed at this step. Loci were defined solely on the basis of chromosomal distance. The defined threshold of 1 Mb was chosen to ensure independence of loci as best as possible. LD structure is later considered by performing conditional analyses within the loci, and, in case of the neighbouring loci 7, 8 and 9, also between loci if considered necessary.

Reviewer #4 (Remarks on code availability):

The github repo has covered the main analyses and are detailedly recorded. We noticed that the authored stated "the in-house annotation pipeline will not be provided". While we understand the privacy of the raw data, would that be possible to list some key analyses during these steps?

Authors response:

We thank the reviewer for code evaluation and the suggestion. We extended the repositories readme to give an overview of the annotations performed by the pipeline.

Reviewer #5

Authors response:

We thank the reviewer for providing valuable comments that helped to improve our work.

Point by point response to reviewer comments

Reviewer #1

This is the second round of revisions. We focussed on the rebuttals provided in the first round of comments.

We appreciate the authors for going through all the comments and addressing them.

Authors' response:

We highly appreciate the constructive feedback. Our responses to the individual comments can be found in the following section.

Comment 1: *The response to comment 5 from reviewer 1:*

One would still need to account for multiple testing when considering a small number of traits. The paper cited <https://doi.org/10.1093/ije/dyr178> compares studies that use this $5e-8$ threshold, but these studies appear to be each focused on a singular trait, hence this citation might not be relevant to the rebuttal argument.

It is not clear how and why the significance threshold is so heavily based on how 'commonly studied' a phenotype is. Further, if the analysis is explanatory then it should be phrased as such. Currently, the abstract does not read that way which can be misleading.

Authors' response:

We thank the reviewer for the attentiveness to statistical rigorousness and agree that multiple testing correction is necessary to control the family-wise error rate (FWER). However, we would like to argue that the choice what constitutes a family of tests can vary depending on the research question. We propose that controlling the FWER for each trait separately or for the whole study can both be justified and that respective findings are interesting for the scientific community. Signals passing the trait-wise genome-wide significance threshold of 5×10^{-8} are of interest for researchers investigating this specific trait. In particular, this holds for GWAS of rare traits for which our knowledge of underlying genetic mechanisms is limited, as it might point towards additional candidate genes, worthwhile to be investigated in further studies. On the other hand, when looking at our study as a whole, controlling the FWER study-wide is more conservative, as correctly pointed out by the reviewer. We believe that including both options provides the highest amount of information to the reader. We agree to improve clarity of the manuscript in this regard.

Changes in manuscript:

We adapted the wording of the abstract, highlighting the exploratory nature of our study and the reliability of signals:

“Smelling is a human sense, expressing strong sexual dimorphisms. We aim to improve the knowledge of the genetics of human olfactory perception by performing an exploratory genome-wide association meta-analysis of up to 21,495 individuals of European ancestry. By sex-stratified and overall analysis of the identification of twelve odours and an identification score, we discovered ten independent loci, seven of them novel, with trait-wise genome-wide significance ($p < 5 \times 10^{-8}$) involving five odours.

Seven of these loci, including four novel ones, are also significant using a stricter study-wide significance threshold ($p < 3.85 \times 10^{-9}$). Loci were predominantly located within clusters of olfactory receptors. Two loci were female-specific while one was sex-differential with respective candidate genes containing androgen response elements. Two-sample Mendelian randomization was applied to search for causal relationships between sex hormones, odour identification and neurodegenerative diseases. A causal negative effect was detected for Alzheimer's disease on the identification score. These findings deepen our understanding of the genetic basis of olfactory perception and its interaction with sex, prioritizing mechanisms for further molecular research."

We explicitly mention how many loci are significant at each threshold (Results – Genome-wide association meta-analysis, 3rd paragraph):

"According to our locus definition (see section 'Locus definition' in Methods), eleven loci [authors comment: ten independent variants, see manuscript] with genome-wide significant associations ($p < 5 \times 10^{-8}$) were detected (see Figure 1) when controlling family-wise error rate (FWER) for each trait, involving associations of five of the twelve odour traits tested as well as the identification score. Seven of these loci (loci 2, 4, 5, 6, 8, 9, 11) remain significant, when applying a stricter threshold of $p < 3.85 \times 10^{-9}$, controlling FWER on a study-wide level by adjusting for multiple-testing across 13 odour traits."

For loci that do not achieve significance when correcting for the number of traits, we mention this fact explicitly in the locus description (Results - Single locus results):

Locus 1: "The index variant achieved trait-wise but not study-wide genome-wide significance."

Locus 3: "The index variant of locus 3 (5q21.3), rs17161232, was associated with lemon identification in the overall analysis ($p = 3.38 \times 10^{-8}$), achieving trait-wise but not study-wide genome-wide significance."

Locus 10: "The index variant showed trait-wise but not study-wide genome-wide significance."

We refined the results summary in the discussion (Discussion, 1st paragraph):

"We identified ten independent loci with trait-wise genome-wide significance ($p < 5 \times 10^{-8}$) of which seven (loci 1, 2, 3, 4, 8, 10, 11) were not previously reported for olfactory traits. Seven loci, including four novel ones (loci 2, 4, 8, 11), further achieved study-wide significance ($p < 3.85 \times 10^{-9}$)."

We changed the wording of the methodology (Methods – Locus definition):

"Variants with $p < 5 \times 10^{-8}$ were considered genome-wide significant, which is consistent with controlling the FWER for each trait separately. We also considered a stricter threshold of $p < 3.85 \times 10^{-9}$, calculated with Bonferroni correction to control the FWER study-wide by additionally accounting for the testing of 13 odour traits."

The legend for table 1 was refined:

"Table 1: Loci with trait-wise genome-wide significance that are associated with olfactory identification. Association statistics and two-sided p-values of β -coefficients of the additive genetic effects are provided for the strongest association of each locus (index variant, minimum p-value). The top association column reports the combination of trait and analysis group (all, male, female) corresponding to this association. Novel loci are presented in bold while significant sex interactions are underlined. Italicized loci fulfil a stricter study-wide significance threshold of $p < 3.85 \times 10^{-9}$, resulting from additional Bonferroni correction on the 13 investigated traits. For clusters of olfactory receptors, the most abundant OR family is provided in parentheses. For independent loci, the q-values (FDR-corrected p-values) for SNP-by-sex interaction of the index variant are shown. *Locus 7 was identified as not independent of locus 8 after locus definition. AA = alternative allele, EA = effect allele."

Further, to comply with the request of reviewer #3, we added the information which loci pass the Bonferroni-corrected threshold to Supplementary Tables S5, S8 and S9, as well as Supplementary Figures S3 and S4.

Comment 2: response to Line 524

Even though the participants may not have reported relatedness, one wouldn't know unless one accounts (tests or controls) for it. Since the authors state in the rebuttal that ancestry is self-reported, it might be worthwhile to have a quick check.

Authors' response:

We agree that relatedness is of concern because it could cause stratification bias. To check this potential issue, we performed LD-score regression for the individual studies. Highest intercept values were observed for the CHRIS study, which however, was already corrected for genetic relatedness. Intercepts of other studies were < 1.06, indicating that there is no substantial bias. We conclude that accounting for relatedness was necessary and done for CHRIS but not for the other studies.

To analyse the impact of ancestry across studies, we also performed MR-MEGA regression for the best-associated traits. We found nominally significant ancestry-related heterogeneity for the index variants of loci 5, 7 (found to be dependent on locus 8) and 9 but all three variants kept their genome-wide significance in the MR-MEGA analysis. We therefore conclude that there is no significant impact of study-wise differing ancestry on our meta-analysis.

Changes in manuscript:

We added the methodology for the computation of the LD-score regression intercepts (Methods – Association analysis, 2nd paragraph):

“For each study, LD score regression intercepts were calculated for each trait and subgroup with LDSC (v. 1.0.1)⁸⁰ to investigate the presence of stratification bias. Default quality criteria of the tool were used for the inclusion of variants. LD scores for European populations provided by the authors of the tool were used as reference. The largest intercepts were observed for the CHRIS study but all were < 1.1. Intercepts of the other studies were < 1.06, indicating that there is no relevant stratification bias (see Supplementary Table S26).”

We added the methodology for MR-MEGA regression (Methods – Analysis of heterogeneity, 2nd paragraph):

“To investigate the impact of study heterogeneity caused by differing ancestries, MR-MEGA regression (v.0.2)⁸⁶ was performed. Variants were not quality-filtered prior to this analysis to prevent the filtering of index variants with low MAF or imputation quality in a single study. The number of principal components used for regression was set to one, the maximal allowed value for an analysis of four studies. P-values for ancestry-related heterogeneity were looked-up for the index variants in their respective best-associated trait.”

We added the result of MR-MEGA analysis (Results – Genome-wide association meta-analysis, 3rd paragraph):

“MR-MEGA revealed nominally significant ancestry-related heterogeneity for index variants of loci 5, 7 and 9, but genome-wide significance was not affected (see Supplementary Table S5).”

Comment 3: please clarify further what differences in cohorts haven't been accounted for

Authors' response:

We thank the reviewer for raising an important point about the limitations of the fixed-effect model used in our meta-analysis. Between-study heterogeneity can potentially bias effect estimates, as it constitutes a violation of the fixed-effect model's assumption of a homogeneous true effect across studies. To account for this, variants with increased I^2 -statistics were removed during quality control. Further, we included known confounders as covariates in the regression models of the single studies and performed MR-MEGA analysis to analyse the impact of ancestry. Meta-regression would allow to explicitly model study heterogeneity in covariables, and with it, an investigation of other potential sources of heterogeneity. However, the number of participating studies in our analysis is too small for such an approach.

Changes in manuscript:

We expanded the limitations section to highlight the limits of the fixed-effect model (Discussion, 9th paragraph):

"These interactions might result in between-study heterogeneity, violating the assumption of our fixed-effect meta-analysis. Modelling heterogeneity with meta-regression techniques considering structural differences between cohorts could therefore be an interesting alternative if the number of studies and their heterogeneity increases in the future."

Reviewer #3

I appreciate the authors' thorough responses to my comments and the revisions made to the manuscript. My concerns have been adequately addressed, and I am generally satisfied with the improvements. I only have two minor comments:

Authors' response:

We thank the reviewer very much for this positive evaluation. Please find below our answers to the individual comments.

Comment 1: *A reference to Table 1 is missing in the main text.*

Changes in manuscript:

A reference to Table 1 was added to the 3rd paragraph in the results section.

Comment 2: *As in Table 1, please also include information on which loci pass the Bonferroni-corrected significance threshold (adjusted for the number of analyzed traits) in the tables and figures of the supplementary material (e.g., Supplementary Figures S3, S4).*

Changes in manuscript:

We added the requested information to Supplementary Tables S5, S8 and S9, as well as Supplementary Figures S3 and S4 and changed legends accordingly.

Reviewer #4

The authors have addressed my and other reviewers' comments and the manuscript has been substantially improved.

Authors' response:

We sincerely appreciate the reviewer's positive evaluation. Below, we provide our responses to the individual comments.

Comment 1: *The only remaining concern is that none of diseases tested showed a nominally significant genetic correlation with the odour identification score. The authors proposed the negative results could be due to small heritability estimates of the identification score. I was wondering if the authors have tested other traits with higher heritability such as pineapple identification in females ($h^2 = 0.12$)?*

Authors' response:

This is a great suggestion and we have therefore expanded our analysis of genetic correlation towards pineapple identification in females using sex-stratified summary statistics of diseases. However, we were still unable to detect nominally significant genetic correlations with complex diseases. It should be noted that, although heritability of pineapple identification in females is higher than for the identification score in the overall analysis, the corresponding Z-scores of the heritability estimate ($Z_{\text{score_all}} = 1.53$, $Z_{\text{pineapple_female}} = 1.54$) are substantially lower than the recommended threshold for genetic correlation analyses of $Z > 4$ (<https://doi.org/10.1093/bioinformatics/btw613>). We therefore also tested coffee identification in the combined analysis, as this trait showed the highest Z-score ($Z_{\text{coffee_all}} = 2.81$). We observed a single nominally significant genetic correlation between this olfactory trait and coronary atherosclerosis ($p = 0.002$, $r_g = 0.334$), although this result does not withstand FDR correction on the number of tested combinations ($q = 0.085$). This demonstrates that the lack of genetic correlation between olfactory traits and complex diseases might be caused by low heritability and/or low sample sizes. Larger studies are required to clarify this issue.

Changes in manuscript:

We updated the methodology (Methods – Genetic correlation with other phenotypes, 1st paragraph): “We screened for genetic correlations between common complex diseases and the overall identification score, coffee identification in the overall analysis and pineapple identification in females. The latter two traits were selected for highest Z-score respectively highest heritability estimate. Summary statistics were retrieved for AD, asthma, coronary atherosclerosis, depression, hypertension, inflammatory bowel disease, multiple sclerosis, obesity, osteoarthritis, osteoporosis, PD, rheumatoid arthritis, stroke, type 1 diabetes and type 2 diabetes from the pan-UKBB website¹⁰⁶. For the genetic correlation with pineapple identification in females, sex-stratified summary statistics of these diseases were used, which however, were not available for AD and inflammatory bowel disease. Genetic correlations were estimated with LDSC (v. 1.0.1)^{80,107}, using the provided LD score reference for European populations. FDR correction was performed for the number of tested phenotype pairs.”

We updated the results of this analysis (Results – Two-sample Mendelian randomization, 6th paragraph):

“We searched for additional phenotypes with possible connection to olfaction by analysing genetic correlations between olfactory phenotypes and complex diseases. We tested the identification score in the overall analysis, as general olfaction was biologically most plausible to correlate with diseases, pineapple identification in females, the trait with highest heritability, and coffee identification in the overall analysis because this trait showed the highest Z-score of the heritability estimate. After multiple

testing-correction, no significant genetic correlations between the three olfactory traits and the considered complex diseases were observed (see Supplementary Table S25).”

Comment 2: Also, for coffee identification in females ($h^2 = 0.09$), has it been tested against coffee or tea consumption in UKB? These data should be available in https://yanglab.westlake.edu.cn/pub_data.html

Authors’ response:

We agree that this is an interesting analysis. As the referenced data is not sex-stratified, we instead tested coffee identification in the overall analysis. We did not observe nominally significant correlations ($p_{\text{intake_coffee}} = 0.076$, $p_{\text{intake_tea}} = 0.782$).

Changes in manuscript:

We expanded our methodology (Methods – Genetic correlation with other phenotypes, 2nd paragraph):

“We further investigated genetic correlation between coffee odour identification and the intake of coffee and tea. Summary statistics for coffee intake and tea intake for individuals with European ancestry and were taken from Xue *et al*¹⁰⁸.”

We added the results of this analysis to the manuscript (Single locus results – Novel loci, Locus 8):
“However, we did not observe associations with coffee identification, indicating that variants reported for coffee consumption might play a stronger role in taste than in smell perception. This is supported by the lack of nominally significant genetic correlation between coffee identification in the overall analysis and the intake of coffee or tea (see Supplementary Table S12).”